# Miniaturization during a Silurian environmental crisis generated the modern brittle star body plan

Ben Thuy [1✉], Mats E. Eriksson[2], Manfred Kutscher[3], Johan Lindgren[2], Lea D. Numberger-Thuy[1] & David F. Wright [4,5]

Pivotal anatomical innovations often seem to appear by chance when viewed through the lens of the fossil record. As a consequence, specific driving forces behind the origination of major organismal clades generally remain speculative. Here, we present a rare exception to this axiom by constraining the appearance of a diverse animal group (the living Ophiuroidea) to a single speciation event rather than hypothetical ancestors. Fossils belonging to a new pair of temporally consecutive species of brittle stars (*Ophiopetagno paicei* gen. et sp. nov. and *Muldaster haakei* gen. et sp. nov.) from the Silurian (444–419 Mya) of Sweden reveal a process of miniaturization that temporally coincides with a global extinction and environmental perturbation known as the Mulde Event. The reduction in size from *O. paicei* to *M. haakei* forced a structural simplification of the ophiuroid skeleton through ontogenetic retention of juvenile traits, thereby generating the modern brittle star bauplan.

[1] Natural History Museum Luxembourg, Department of palaeontology, 25, rue Münster, 2160 Luxembourg, Luxembourg. [2] Department of Geology, Lund University, Sölvegatan 12, SE-223 62 Lund, Sweden. [3] Dorfstrasse 10, 18546 Sassnitz, Germany. [4] Department of Paleobiology, National Museum of Natural History, Smithsonian Institution, Washington, DC 20013–7012, USA. [5] American Museum of Natural History, Division of Paleontology, Central Park West at 79th St, New York, NY 10024, USA. ✉email: bthuy@mnhn.lu

The origin of major organismal clades is among the most debated subjects in the fields of palaeontology and evolutionary biology. Although molecular clock estimates have greatly improved the accuracy of evolutionary trees, basal divergences still are notoriously difficult to constrain temporally[1,2]. Even in those cases where fossil-calibrated molecular clock estimates converge to a narrow time interval[3], direct ancestral representatives are almost never known from fossil evidence[1]. As a consequence, inferences on correlations between divergence times and other large-scale events in Earth history, e.g., mass extinctions or global climate perturbations, are largely hypothetical. Here, we present an exceptional case where the origin of a major modern animal clade can be directly pinpointed to a specific pair of temporally consecutive species that occurred during a well-constrained time interval coinciding with major environmental and faunal reorganizations.

Ophiuroids (brittle stars), the slender-armed "cousins" of starfish, are major components of modern marine benthos. They occur in all of the world's oceans[4], making them excellent model organisms with which to explore macroevolutionary patterns[5]. In addition, they have a rich fossil record; although articulated skeletons are known only from a handful of localities (Lagerstätten), dissociated skeletal plates occur abundantly in shallow-marine strata as microscopic fossils[6]. Some of these remains, particularly the spine-bearing lateral arm plates, are identifiable to species level and thus can be used in phylogenetic studies[7,8]. Modern brittle stars all belong to the subclass Myophiuroidea[9], the sole surviving clade of ophiuroids that radiated during the Early Paleozoic, between 480 and 420 Mya. In spite of considerable progress in deciphering the phylogenetic relationships of living ophiuroids[8,10], their deep time origin remains elusive.

Here, we describe two new fossil taxa from Silurian strata on the island of Gotland, Sweden, placing them in a phylogenetic context. Furthermore, we analyze the stratigraphic succession and associated ophiuroid microfossil assemblages, that collectively consisted of more than 1300 specimens, (Fig. 1; Supplementary Figs. 1, 2), that range in age from the Telychian (latest Llandovery, about 433 Mya) to Ludfordian (late Ludlow, about 423 Mya)[11,12]. The microfossil material includes two taxa with lateral arm plates and vertebrae that show a unique combination of ancestral and derived characters, the latter of which are previously known only from considerably younger taxa[13].

## Results and discussion
### Systematic palaeontology. Ophiuroidea Gray, 1840

*Ophiopetagno paicei* gen. et sp. nov. Thuy, Eriksson & Numberger-Thuy

Etymology: The generic name honours "heavy metal painter" Joe Petagno in recognition of his artistic talent and ability to integrate palaeontological imagery into his art pieces, to the enrichment of both the music scene and the world of science; the species name honours Ian A. Paice, drummer of legendary rock band Deep Purple, one of the last common ancestors in heavy metal.

Holotype: Natural History Museum Luxembourg (MnhnL) OPH087, a lateral arm plate.

Referred material: Ninety-six lateral arm plates and eight vertebrae (OPH092-OPH093; OPH101-OPH105).

Locality and horizon: Trench exposure 300 m south of Klintehamn on Gotland, Sweden; locality called Svarvare; Gannarve or Svarvare member (extinction phase of the Mulde Event), *Ozarkodina bohemica longa* conodont Zone, Fröjel Formation, Slite Group, Whitwell Stage, Wenlock Series, 428.5 ± 0.7 Mya.

Description: Lateral arm plates (Fig. 1; Supplementary Fig. 3) small, elongate, with dorsal edge slightly concave due to weak constriction. Ventral plate portion protruding and distally incised by tentacle notch. Outer surface stereom finely meshed with trabecular intersections transformed into small tubercles; vertical band of more fine-meshed stereom along distal edge of plates, devoid of spurs but with poorly defined furrow parallel to proximal plate edge. Up to three lateral arm spine articulations in vertical row, each consisting of single opening surrounded by elevated ridge. Inner side of lateral arm plates with large, sub-triangular vertebral articulation; ventral edge of lateral arm plates slightly convex and lined with up to five bulges each showing small perforation on inner side, corresponding to groove spine articulations. Vertebrae (Fig. 1; Supplementary Fig 3) composed of fused ambulacrals, roughly cylindrical, with slightly concave waist; lateral sides with deep podial basin distally, and sub-triangular articulation surface proximally, matching vertebral articulation on inner side of lateral arm plates. Distal side with large, dorsally converging zygocondyles; water vessel canal running entirely within vertebra, distally enclosed by large zygosphene.

*Muldaster haakei* gen. et sp. nov. Thuy, Eriksson & Numberger-Thuy

Etymology: The generic name derives from *Mulde*, after both the type locality and the eponymous palaeoenvironmental event, and *astér* (Greek) meaning star, a common suffix in asterozoan names; the species epithet honours Tomas Haake, Swedish drummer of metal band Meshuggah, representing a derived form of heavy metal music.

Holotype: Natural History Museum Luxembourg (MnhnL) OPH088, a dissociated lateral arm plate.

Referred material: Two hundred ninety-seven lateral arm plates, six vertebrae, four articulated arm fragments and two putative predation regurgitates (Supplementary Fig. 4) or fecal pellets composed of dissociated lateral arm plates and vertebrae (OPH089-OPH091; OPH094-OPH100).

Locality and horizon: Former brick-yard near Mulde; "Mulde Brick clay" type locality, on Gotland, Sweden. *Gothograptus nassa-Pristiograptus dubius* graptolite Interregnum, Mulde Event, fauna 4 (survival phase of the Mulde Event), Halla Formation, Slite Group, Gleedon Stage, Wenlock Series, 428.1 ± 0.7 Mya[11].

Description: Lateral arm plates (Fig. 1; Supplementary Fig. 4) very small, strongly elongate and weakly constricted. Ventral portion protruding and distally incised by tentacle notch. Outer surface stereom coarsely meshed, with trabecular intersections transformed into irregular tubercles; vertical band of much more finely meshed stereom lining proximal edge, with furrow parallel to proximal plate edge. Two closely spaced lateral arm spine articulations each composed of single opening surrounded by elevated ridge. Inner side of lateral arm plates with elongate vertebral articulation. Ventral edge of lateral arm plates slightly convex, lacking bulges, perforations and other traces of groove spine articulations. Vertebrae (Fig. 1; Supplementary Fig. 4) bone shaped, with strongly concave waist. Lateral sides with deep podial basin distally, and elongate, poorly defined articulation surface proximally matching vertebral articulation on inner side of lateral arm plate. Large, coarsely tuberculated zygocondyles on distal side of vertebrae merged into hourglass-shaped knob. Zygosphene composed of two widely separated, oblique knobs at ventral tips of zygocondyles. Water vessel canal enclosed by distal third and proximal quarter of vertebra, exposed at waist of vertebral body running in shallow furrow. Articulated arm segments with lateral arm plates meeting along ventral midline, exposing podial basins ventrally and vertebrae dorsally. No trace of dorsal or ventral arm plates. Short, conical, pointed lateral arm spines.

### Systematic affinities of the new fossils. *Ophiopetagno paicei* is morphologically intermediate between Paleozoic and modern ophiuroids. The groove spines and fully enclosed water vessel canal are ancestral characters placing *O. paicei* close to Paleozoic

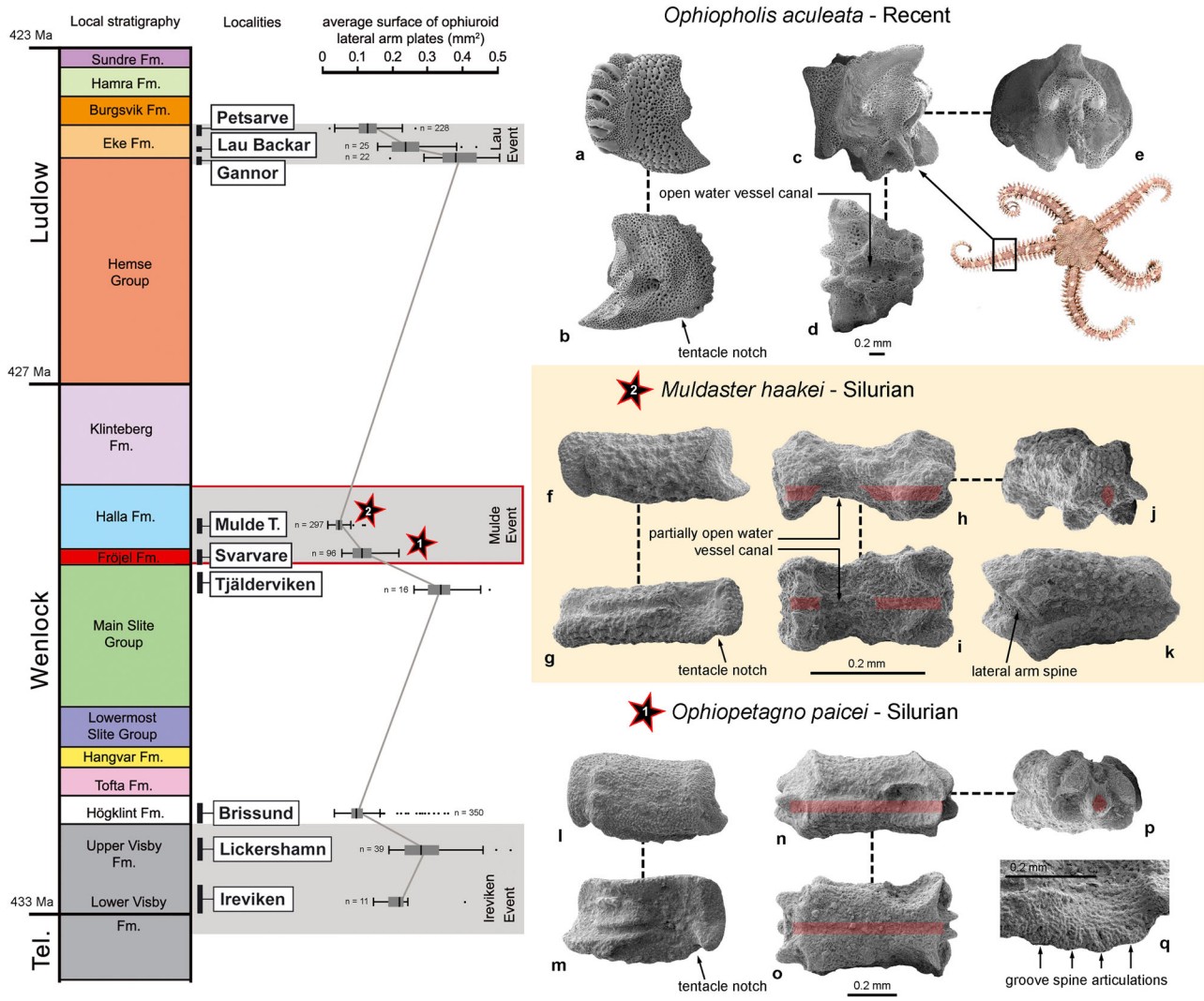

**Fig. 1 Arm plates of *Ophiopetagno paicei* and *Muldaster haakei* from the Silurian of Gotland, Sweden, in comparison to a living relative (*Ophiopholis aculeata*), and in the context of the local stratigraphic framework[12] with localities sampled (Mulde T.: Mulde Tegelbruk), palaeoenvironmental events, and trends in average surface area of the lateral arm plates as an approximation for body size.** *Ophiopholis aculeata*: lateral arm plate in external (**a**) and internal (**b**) views. Vertebra in lateral (**c**), ventral (**d**), and distal (**e**) views. *Muldaster haakei*: Specimen MnhnL OPH088 (all numerical codes in the figure legend refer to specimen numbers in the MnhnL collection), holotype of *M. haakei*. Lateral arm plate in external (**f**) and internal (**g**) views. Specimen OPH089. Vertebra in lateral (**h**), ventral (**i**), and distal (**j**) views. **k** Specimen OPH090. Arm segment in ventral view. *Ophiopetagno paicei*: Specimen OPH087. Lateral arm plate in external (**l**) and internal (**m**) views. Specimen OPH092. Vertebra in lateral (**n**), ventral (**o**), and distal (**p**) views. **q** Specimen OPH93. Lateral arm plate, detail of tentacle notch in internal view.

stem taxa with ambulacral halves fused into vertebrae, in particular *Hallaster*, from the Ordovician Period, and the Silurian genera *Lapworthura* and *Furcaster*[14,15]. On the other hand, the lateral arm plates of *O. paicei* have a distinct ventral portion incised by a tentacle notch, a derived character otherwise found exclusively in the clade including the modern ophiuroids.

*Muldaster haakei* shares many features with *O. paicei* regarding the lateral arm plate and vertebral morphologies, but differs in the lack of groove spine articulations and a partially exposed water vessel canal, both of which are derived characters. The geographic proximity and immediate stratigraphic succession suggest that *M. haakei* evolved directly from *O. paicei*. In order to underline the substantial nature of morphological modifications, however, we assign the new taxa to separate genera.

**Phylogenetic relationships.** In order to determine the phylogenetic positions of *O. paicei* and *M. haakei*, we performed a

morphological cladistic analysis with 68 characters using undated and tip-dated Bayesian inference (Fig. 2, Supplementary Methods, Supplementary Fig. 5, Supplementary Data 1). Apart from *O. paicei* and *M. haakei*, our analysis included: (1) representatives of all major Paleozoic ophiuroid groups; (2) members of the two extant superorders Euryophiurida and Ophintegrida; (3) the most exhaustively known Paleozoic modern-type ophiuroids (*Ophiurina lymani*, *O. armoricana*, *Stephanoura belgica*, *Ophiaulax decheni* and *Aganaster gregarius*)[13,15,17–20] suspected to be part of the extant ophiuroid ancestry[16]; and (4) the stenurid *Pradesura jacobi*[15] as outgroup taxon. The resulting tree unambiguously shows that *O. paicei* and *M. haakei* are nested at an early-diverging position of the clade that includes all living ophiuroids (Fig. 2, Supplementary Fig. 5). To our knowledge, this is the first deep-time phylogeny of the Ophiuroidea that includes both extant and Paleozoic clades. Furthermore, in exceptional clarity, it illustrates that evolutionary modifications seen in these Gotland ophiuroids represent the onset of a stepwise acquisition of

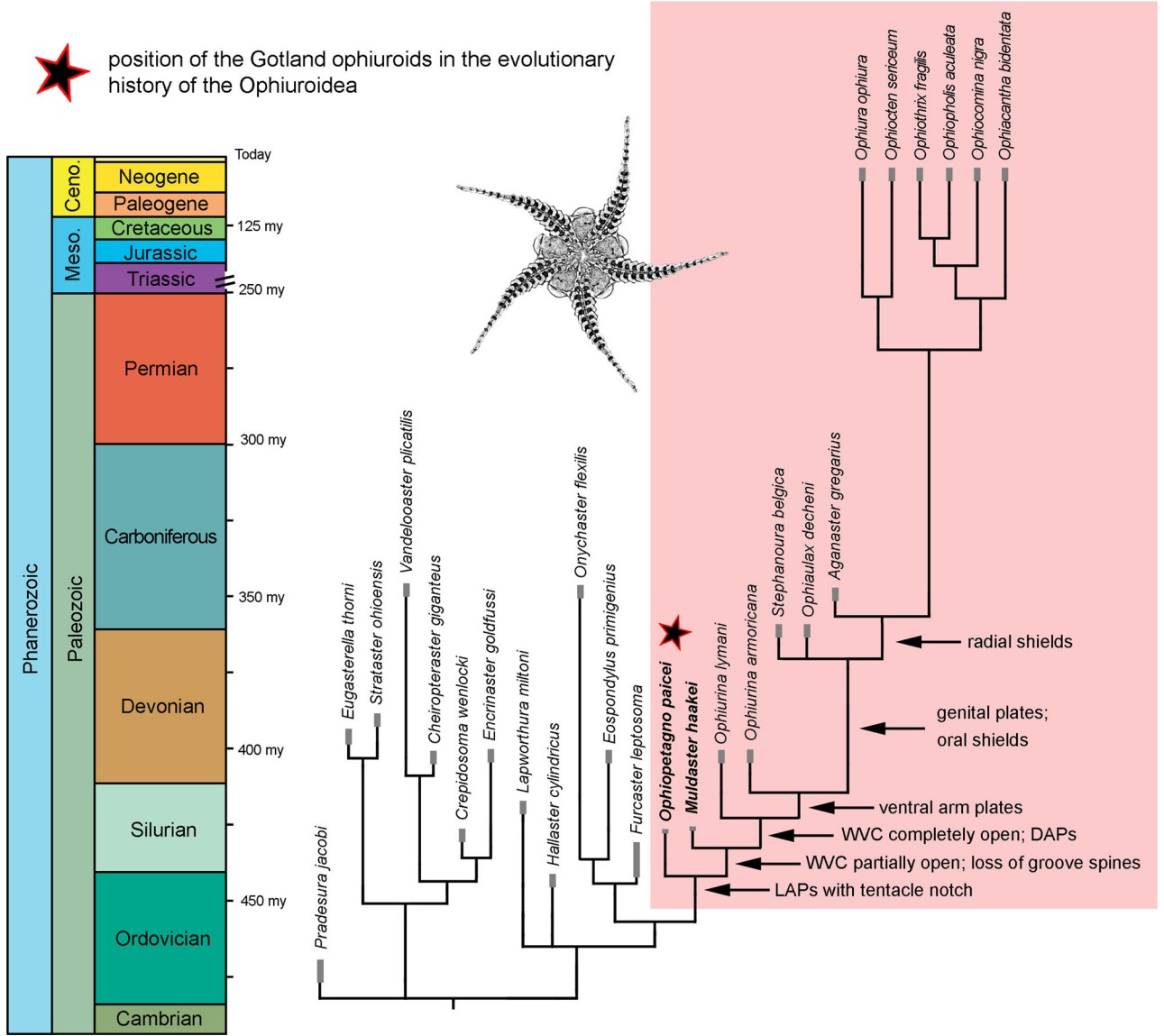

**Fig. 2 Evolutionary tree of the Ophiuroidea to show the position of *Ophiopetagno paicei* and *Muldaster haakei* from the Silurian of Gotland, Sweden.** Arrows indicate synapomorphies at the respective nodes. DAP dorsal arm plate, LAP lateral arm plate, VWC water vessel canal, Ceno. Cenozoic, Meso. Mesozoic. Gray lines indicate documented fossil range of taxa.

synapomorphies along a series of nested stem taxa and that the inferred tree topology is remarkably congruent with their successive stratigraphic age (Fig. 2).

Posterior probability (PP) calculations of ancestor-descendant relationships across the distribution of tip-dated phylogenies confirm with maximum support (PP = 1.00, i.e. ancestor-descendant pairs were recovered in all 1500 trees) that *O. paicei* and *M. haakei* are, indeed, in a direct ancestor-descendant lineage, sampled in a sufficiently complete fossil record (Supplementary Fig. 6). Whether a phylogenetic tree faithfully resolves a direct ancestor-descendant relationship generally depends on scaling. At the resolution normally achievable using fossil evidence, only few extinct side lineages are detected and the tip sampling rate often appears to be high. At a higher resolution, however, many more extinct lineages become visible, and what appeared to be direct ancestors at a lower resolution almost always turn out not to be direct ancestors. Our study seems to be an exception in this respect in that the analyses show the two key species *O. paicei* and *M. haakei* to be direct ancestors in spite of the (comparatively) high resolution. The temporal proximity

between these two species, as well as between *M. haakei* and the remaining crownward clade, strongly suggest that the species pair represents a rare case of a direct ancestor-descendant relationship in the fossil record. Recognizing ancestor-descendant pairs in the fossil record has major implications for discerning alternative modes of speciation in the fossil record, including anagenesis and "budding" cladogenesis[21]. Anagenesis corresponds to a mode of morphospecies origination resulting from phenotypic changes within a single evolving lineage, which predicts a pattern of sequential, non-overlapping stratigraphic ranges between ancestor-descendant pairs. In contrast, speciation via budding cladogenesis involves a lineage splitting event where the ancestral lineage temporally persists beyond the origination of its daughter lineage, and is evidenced by overlapping stratigraphic ranges between ancestor-descendant pairs. Based on our dataset of more than 1000 lateral arm plates from the Silurian of Gotland, we find no evidence *O. paicei* and *M. haakei* temporally overlap, which indicates positive support for anagenesis. However, the hypothesis of budding cladogenesis is difficult to rule out entirely. An incompletely sampled fossil record is biased toward recovering

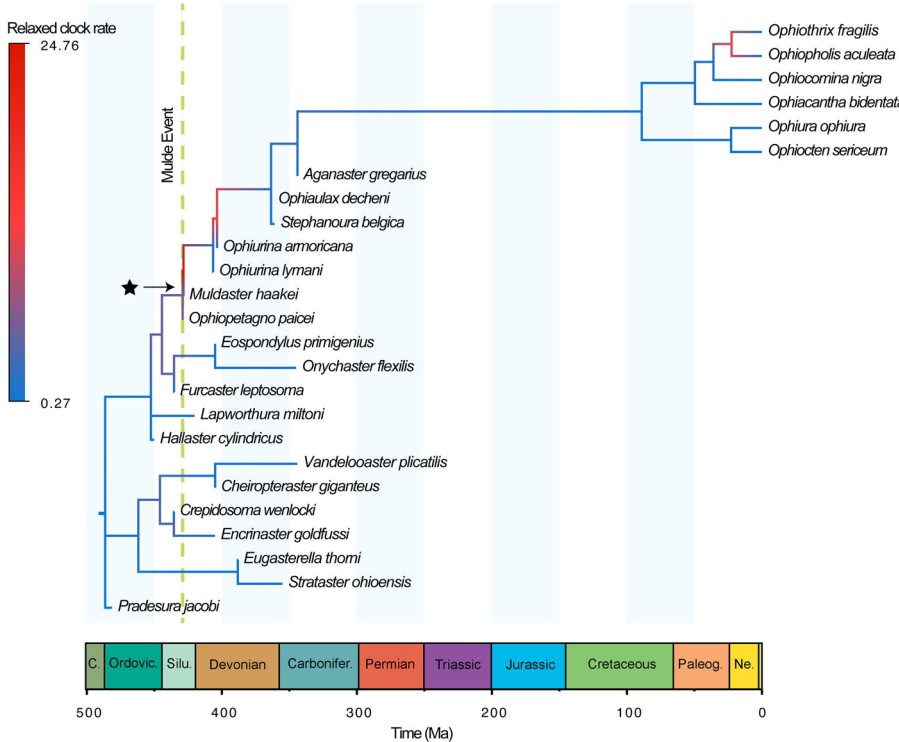

**Fig. 3 Time-calibrated phylogeny from the tip-dating analysis with mean per-branch rates of discrete character evolution mapped to indicate low (blue colors) vs. high rates (red colors) of morphological evolution inferred from a model allowing each branch to have its own rate of change.** The star indicates the branch with the highest rate in the tree (rate multiplier = 24.76), which links *Ophiopetagno paicei* and *Muldaster haakei* and occurs concomitant with the Mulde Event.

anagenesis over budding cladogenesis, and it is plausible that the same rapid environmental changes (and their associated selective pressures) that led to the origination of *M. haakei* may have led to the extinction of *O. paicei* shortly after their lineages diverged. Regardless of how the transition between *O. paicei* and *M. haakei* took place, the uncorrelated relaxed morphological clock model incorporated in the time-calibrated tree analysis indicates that the acquisition of synapomorphies leading to the origin of the modern brittle star body plan was rapid, and that changes within body plan characters during this transition occurred at a rate much higher than background rates of change within the Ophiuroidea (Fig. 3).

**Palaeobiological context.** The transition from *O. paicei* to *M. haakei* is intriguing, both with respect to timing and inferred driving mechanisms. The strata yielding the ophiuroid remains were deposited in a shallow, tropical sea with prominent reef buildups[12,22]. Moreover, the sampled 8-million-year-interval encompasses three global biotic and environmental perturbations known as the Ireviken, Mulde, and Lau events, respectively[22,23]. Fossils assigned to *O. paicei* and *M. haakei* occur in sediments deposited during the extinction and survival phases, respectively, of the Mulde Event, which is also known as the 'Big Crisis' (Fig. 4); that is, one of the most prominent extinction events during the Silurian Period[11,23]. Notably, this event is associated with size decrease, or Lilliput Effects[24], in a number of taxa[11]. Accordingly, we analyzed the body size trend in our ophiuroid assemblages across the sampled interval, using the average surface area of the lateral arm plates as a proxy for body size (Supplementary Data 2, Supplementary Table 1). Our results show that the average size of the ophiuroids substantially decreased during or immediately after each of the three extinctions (Fig. 1), presumably as a response to unfavorable

environmental conditions through shorter developmental times or lower energy requirements[25]. Significance tests using Monte Carlo simulations indicate that the magnitude of body size decrease across all three environmental change events is statistically significant and highly robust to sampling artifacts and incompleteness of the fossil record (Ireviken, $p < 0.0003$, Mulde, $p < 0.0001$, Lau, $p < 0.0001$), thus very likely representing a biological pattern rather than sampling bias alone (Supplementary Fig. 7). The most prominent size decrease coincides temporally with the Mulde Event and corresponds to the first appearance of *M. haakei*—a species with lateral arm plates that are up to four times smaller than other ophiuroids in the sampled material. The possibility that *M. haakei* represents ontogenetically immature individuals is confidently precluded by the presence of fully differentiated stereom with strongly contrasting mesh sizes; these only occur in proximal lateral arm plates of adult animals[7].

Because the diminutive size of *M. haakei* is accompanied by distinct morphological modifications, this process qualifies as miniaturization rather than dwarfing[26]. Likely as a result of this pronounced size decrease, the lateral arm plates of *M. haakei* met ventrally, which forced a structural simplification of the entire arm skeleton, especially noticeable in the loss of the groove spines. However, this re-arrangement also allowed for a partial opening of the water vessel canal. Furthermore, the elongated arm joints, the lateral arm plates meeting dorsally and ventrally, and the low number of lateral arm spines are traits typically found in paedomorphic representatives of the extinct Protasteridae[27] and the living ophiuroids[28]. We therefore interpret the observed miniaturization as being caused by paedomorphosis; that is, the retention of juvenile traits into adulthood.

**Evolutionary implications.** The skeletal modifications observed in the species pair *O. paicei* and *M. haakei* initiated a stepwise

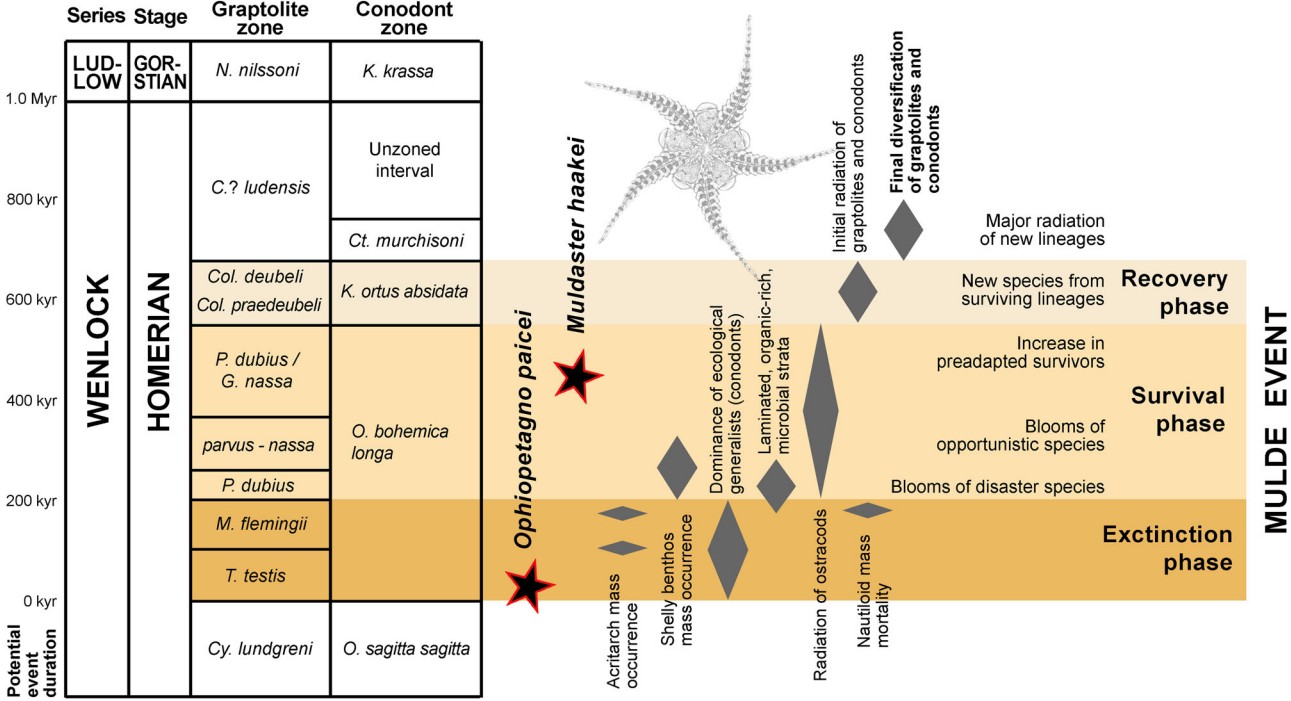

**Fig. 4 Summary diagram of the Mulde Event (the "Big Crisis"[11]) in the Silurian of Gotland, Sweden.** Occurrences of the ophiuroid species *O. paicei* and *M. haakei* and biotic events are plotted.

acquisition of additional synapomorphic traits, in particular dorsal and ventral arm plates, along a succession of stratigraphically consecutive stem taxa (Fig. 2). We hypothesize that this evolutionary trend towards an increased complexity of the modern ophiuroid skeleton paralleled a gradual increase in body size, culminating in the first large-sized modern ophiuroid in the Early Carboniferous[16]. This is one of the exceptionally rare cases where the origin of a major animal clade can be pinpointed precisely in terms of timing, location and driving palaeoecological mechanisms. Environmental perturbations, in particular mass extinctions, have been amply studied as major evolutionary catalysts[29], but understanding of the underlying processes remains vague. Often, the last common ancestor of clades emerging directly from post-extinction radiations remains undetected. In addition, the stratigraphic resolution is generally too low to allow for robust correlations between the temporal succession and pivotal speciation events. Our study therefore provides an exceptional insight into how environmental perturbations can trigger evolutionary innovation, which, in times of growing global habitat deterioration and climate change, is key for a better understanding of biotic responses to environmental stress.

## Methods

**Localities sampled**. Ten localities on Gotland, Sweden, were sampled and processed for ophiuroid microfossils (Fig. 1; Supplementary Figs 1, 2).

Ireviken[30,31]: Llandovery and Wenlock series, Telychian and Sheinwoodian stages, *Pterospathodus amorphognathoides* through Upper *Kockellela ranuliformis* conodont zones, Lower Visby Formation through Högklint Formation (unit a). Ophiuroid sample is from the middle–upper portion of the Lower Visby Formation.

Lickershamn[30,31]: Wenlock Series, Sheinwoodian Stage, Lower *Pterospathodus procerus* through Upper *Kockellela ranuliformis* conodont zones, Lower Visby Formation (unit e) through Högklint Formation (unit b). Ophiuroid sample is from the middle part of the Upper Visby Formation.

Brissund[32,33]: Wenlock Series, Sheinwoodian Stage, upper *Kockellela ranuliformis* conodont Zone, Högklint Formation "undifferentiated".

Tjälderviken (see Tjeldersholm for locality references)[30,31]: Wenlock Series, Whitwell Stage, *Ozarkodina sagitta sagitta* conodont Zone, Slite Group (upper part —"*Pentamerus gothlandicus* beds" or equivalents).

Svarvare[30,31]: Wenlock Series, Whitwell Stage, *Ozarkodina bohemica longa* conodont Zone, Slite Group, Fröjel Formation, Gannarve or Svarvare member.

Mulde Tegelbruk[30,31]: Wenlock Series, Gleedon Stage, *Ozarkodina bohemica longa* conodont Zone, Halla Formation, Mulde Brick clay member, *Gothograptus nassa-Pristioprion dubius* graptolite Interregnum, Mulde Event, fauna 4.

Gannor[30,31]: Ludlow Series, Ludfordian Stage, uppermost part of the *Polygnathoides siluricus* conodont Zone or lowermost part of the Lower Icriodontid conodont subzone *sensu* Jeppsson[34], Hemse Group, När Formation, Botvide Member or the lowermost Eke Formation.

Lau backar[38,39]: Ludlow Series, Ludfordian Stage, uppermost part of the Lower Icriodontid conodont subzone *sensu* Jeppsson[34], Eke Formation.

Petsarve[30,31]: Ludlow Series, Ludfordian Stage, Upper Icriodontid conodont subzone *sensu* Jeppsson[34], Eke Formation, upper part.

Hoburgen[30,31,35]: The Hoburgen area comprises several mapped geological localities spanning the Ludlow Series, Ludfordian Stage, *Ozarkodiana snajdri* through *O. crispa* conodont zones, uppermost Burgsvik Formation through the Sundre Formation. The ophiuroid sample is from weathered material within a cave adjacent to the so-called Hoburgsgubben ("Hoburg man") sea stack; Hamra or Sundre Formation.

In all the samples, the ophiuroid material was fully disarticulated, with very few exceptions from the Mulde Tegelbruk sample (Fig. 1, Supplementary Fig. 4). Furthermore, we assessed the existence of pre-burial sorting in the samples by semi-quantitative examination of the plate type proportions for various asterozoan and pelmatozoan echinoderm groups. In all cases, relative abundances of skeletal elements reflected their expected anatomical frequency irrespective of their hydrodynamic properties (shape, size). The plate types that are the most abundant in the skeleton of a given echinoderm taxon were also the most abundant in the samples, and no particular plate type was over- or underrepresented with respect to their anatomical frequency. We therefore exclude the possibility that the studies skeletal plates underwent pre-burial sorting.

**Geological context and event stratigraphy**. The island of Gotland, Sweden, exposes stacked generations of carbonate platforms formed in shallow marine settings along the northwestern margin of the intracratonic Baltic basin during the Silurian Period[12,36,37]. The rock succession is characterized by a nearly complete absence of tectonic overprinting and only minor dolomitization occurs. These characteristics combined with a palaeolatitudinal setting of the island of c. 20° south of the equator, have resulted in rich, diverse and exceptionally preserved tropical fossil faunas. The entire succession is sub-divided into a number of formations and groups, ranging from the upper Llandovery Lower Visby Formation through the upper Ludlow Sundre Formation (Fig. 1; Supplementary Fig. 2)[22,36]. The facies belts change along outcrop strike, from proximally formed limestones in the northeast to sparsely graptolitic, more distally formed marls in the southwest. The stratigraphic completeness combined with excellent fossil preservation, and

the long history of geological investigations has produced a very high-resolution biostratigraphic framework, primarily based on conodonts[22].

During the last few decades, the Silurian Period has emerged as a highly dynamic and climatically unstable time during Earth History, encompassing a whole suite of biogeochemical events[38,7–40]. In the Gotland succession alone, high-resolution studies primarily based on conodonts enabled the identification of up to eight such events; the most prominent ones being the Ireviken Event, Mulde Event, and Lau Event (Fig. 1)[12,22,23,41,42]. These latter events, characterized by rapid and profound shifts in the oxygen and carbon isotope signatures, as well as faunal and sedimentary facies restructurings, have been shown to have a very widespread, if not global, geographic occurrence[22,23,39].

The Homerian (middle Silurian) Mulde Event (Fig. 4), which is of particular interest in this study, was first identified on Gotland[42], based on conodont fauna re-organizations. However, prior to this a major Homerian extinction event among graptolites had been noted in other parts of the world, with as much as 95% of all species going extinct in some regions[11,39]. Therefore, parts of the Mulde Event are known also under different ("graptolite-based") names, such as the "*lundgreni* Event" and the "Big Crisis"[12], the latter being a term recently used "to refer to the sum total of all global change during this interval, not only the event as it pertains to graptolites"[11].

On Gotland, the Mulde Event begins at the base of the Fröjel Formation and ends at the top of the Halla Formation (Fig. 4). In addition to stepwise extinctions of graptolites and conodonts (as well as other taxa that have not been as intensely studied) during three datum points (Datum 1. 1.5. and 2), the Mulde Event is characterized by sedimentary changes and major stable isotope perturbations recorded on widely separated continents[39]. The triggering mechanism/s of the event have been debated, but are generally thought to be linked to the relationship between, and changes in, ocean circulation, primary production, and the global carbon cycle[23,39,41,43].

**Sample treatment**. Bulk sediment samples taken by one of us (M.K.) were screen-washed using tap water. Ophiuroid microfossils were picked from dried sieving residues under a dissecting microscope. Selected specimens were cleaned in an ultrasonic bath, mounted on aluminum stubs and gold-coated for scanning electron microscopy (SEM) using a Jeol Neoscope JCM-5000. Skeletal plates of the recent ophiuroid *Ophiopholis aculeata* used for comparison with the Gotland ophiuroids were extracted from an articulated individual using household bleach, rinsed in tap water and mounted for SEM[8]. All figured specimens are deposited in the collections of the Natural History Museum Luxembourg (MnhnL).

**Nomenclatural Acts**. This published work and the nomenclatural acts it contains have been registered in ZooBank, the proposed online registration system for the International Code of Zoological Nomenclature (ICZN). The ZooBank LSIDs (Life Science Identifiers) can be resolved and the associated information viewed through any standard web browser by appending the LSID to the prefix "http://zoobank.org/". The LSIDs for this publication are: urn:lsid:zoobank.org:pub:0E3C9E87-C337-4509-B02C-6053492B814B, for the publication, urn:lsid:zoobank.org:act:E378D576-4B79-45CC-A9B3-C46999D1A753 for *Muldaster*, urn:lsid:zoobank.org:act:01ED8A54-507D-4709-8E38-E841C021F879 for *M. haakei*, urn:lsid:zoobank.org:act:395AAF1A-087F-4897-9B2C-CE0132284EDC for *Ophiopetagno*, and urn:lsid:zoobank.org:act:A705CAD9-B96B-448C-A0C7-5C293D03ABB5 for *O. paicei*.

**Phylogenetic analysis**

*Bayesian inference analysis*. The phylogenetic position of *Ophiopetagno paicei* and *Muldaster haakei* was evaluated with Bayesian-inference analyses applied to a data matrix of 25 taxa and 68 characters (Supplementary Methods, Supplementary Data 1). The set of taxa includes representatives of all major Paleozoic ophiuroid groups, six living species representing the three major extant ophiuroid clades, and the best known Paleozoic modern-type ophiuroid species. The outgroup taxon was chosen because the Stenurida are widely accepted as sister to the Ophiuroidea, irrespective of whether they represent a class of their own or a paraphyletic complex at the base of the Ophiuroidea[14]. *Pradesura jacobi* was chosen as one of the best known stenurid taxa.

All characters were unordered in the absence of evidence for a clear ontogenetic or size-related progression between character states. Scoring was carried out independently by two of us (BT and LNT) and then compared and checked for consistency. The only inconsistency pertained to character 48, resulting from an uncertain interpretation of the presence or absence of a constriction in the lateral arm plates of some Paleozoic taxa. Consistency of scoring was restored through a case-by-case revision of the concerned taxa, rigorously applying the definition of constriction in lateral arm plates as previously provided[7,8]. When character states could not be assessed due to poor preservation or lack of data, the character was scored with a "?". If a character was inapplicable in a taxon, e.g. pertaining to a structure which was absent in that particular taxon, the character was scored with a "-".We used the software Xper2[44] to assemble our matrix.

Bayesian inference was performed with MrBayes[44] using MCMC with default parameters for morphological data[8,46]. MrBayes uses a modified version of the Juke-Cantor model for morphological data for binary and multi-state

characters[45,46]. Only variable characters were sampled, omitting characters that have the same state for all examined taxa, and we compensated for character selection bias[46]. We assumed that all character states have equal frequency, that prior probabilities were equal for all trees, and that evolutionary rates vary between sites according to a discrete gamma distribution. Branch lengths were estimated using compound Dirichlet priors, which first specify a tree length using a gamma distribution and subsequently uses a Dirichlet distribution to partition the tree length into branch lengths[47]. Additional analyses indicate the tree topology was unaffected by the number of chains (four vs. eight) and changes in temperature of the MCMC analysis. It has been shown that likelihood-based methods such as Bayesian statistics outperform parsimony under these circumstances and produce more reliable tree hypotheses[46]. Trees (and other parameters of the model) are sampled from the corresponding marginal PP distributions. We summarized the posterior distribution of trees by computing a majority rule consensus tree. Average standard deviations of split frequencies stabilized at about 0.007–0.01 after 3 million generations, sampled every 1000 generations. The first 25% of the trees were discarded as burnin. The consensus trees were examined with the software FigTree v. 1.4.2 by Rambaut (http://tree.bio.ed.ac.uk/software/figtree/). As is common standard in statistics, we regard Bayesian credibility intervals of 95–99% as strong support, and at least 90% probability as good support (Supplementary Fig. 5).

*Tip-dated phylogenetic analysis and rates of discrete character evolution*. A tip-dating analysis using the sampled-ancestor implementation of the fossilized birth death process (SA-FBD)[48–51] was conducted to obtain a posterior distribution of time-calibrated phylogenies used for downstream analysis of macroevolutionary inferences. The analysis used the same character matrix as the undated phylogenetic analyses described above.

As described for the undated analysis, we used the Mk model of character evolution[52]. Rate variation among characters was modeled using a lognormal distribution[53]. Rate variation among lineages was modeled using an independent gamma rates (IGR) clock model with a broad uniform prior spanning multiple orders of magnitude. We used a time-varying implementation of the SA-FBD model where macroevolutionary and sampling parameters are estimated for each geological stage, resulting in a time-series of speciation ($\lambda$), extinction ($\mu$), and fossil sampling and recovery ($\psi$) rates[54–56]. A wide, uniform prior distribution spanning the middle Cambrian to Ordovician was placed on the age of the most recent common ancestor of the clade (i.e., the tree age). Additional details concerning the choice of prior distributions used are discussed in the Supplementary Methods. Markov-chain Monte Carlo analysis was performed in MrBayes for 10 million generations with the first 25% of samples discarded as burn-in. Estimated sample sizes for all parameters were >300, with most >1000, and the potential scale reduction factor (PSRF) was ~1.00 for all parameters. The total number of post-burnin time-calibrated trees used in downstream analyses was 1500.

The relaxed clock model of morphological evolution employed in the tip-dating analysis was simultaneously used to estimate the mean per-branch rate of discrete character evolution across the phylogeny (see Wright[55] for a similar analysis in fossil crinoids and featuring a detailed description of methods). A time-calibrated maximum clade credibility tree with the distribution of mean rates among branches is shown in Fig. 3.

*Completeness of the ophiuroid fossil record and sampling ancestor-descendant relationships*. The completeness of the fossil record[57] can be quantified as the proportion of extinct taxa with a fossil record that has been sampled at least once. Because our study involves a possible case documenting evolutionary change spanning a speciation event (or sequence of speciation events) in the fossil record, we explored the completeness of the ophiuroid fossil record over time and tested whether sampling ancestor-descendant pairs should be expected from that record, particularly during the Silurian. Formally, completeness can be calculated as:

$$\Phi = \psi/(\mu + \psi) \tag{1}$$

where $\psi$ and $\mu$ are the fossil sampling and extinction rates, respectively (see Eqs. 1a,1b in the appendix of Foote[58]. Using the posterior sample of parameter estimates from the tip-dated analysis, we calculated $\Phi$ for each geological stage using parameters estimated from the SA-FBD model. Completeness metrics for the Ordovician—Permian stages are depicted in Supplementary Fig. 6.

The recovery of sampled ancestors implies either direct anagenetic transformation or "budding" cladogenesis (particularly if species temporal ranges overlap[57,59]). The probability of sampling an ancestor-descendant pair (AD) can be calculated as $\Pr[AD] = \Phi^2$[58].

To evaluate empirical support for whether *Ophiopetagno paicei* and/or *Muldaster haakei* represent sampled ancestors (i.e., fossil morphotaxa that are either directly or indirectly ancestors to another fossil morphotaxon[57]), we calculated the PP of ancestor-descendant relationships by dividing the frequency in which a morphotaxon was placed as an ancestor in the posterior distribution of time-calibrated trees by the total number of trees sampled ($N = 1500$ phylogenies).

*Sensitivity analyses of phylogenetic and macroevolutionary inferences*. To evaluate whether our phylogenetic and macroevolutionary inferences are sensitive to variation in assumptions and choice of prior distributions, we conducted a series

of six additional tip-dated analyses that vary in choice of one or more of the following: FBD priors, constant vs. time heterogeneous rates of sampling and diversification, and clock model priors (Supplementary Methods, Supplementary Table 2). In particular, we explored whether our results are robust to alternative priors and assumptions regarding fossil sampling. Careful attention to fossil sampling is necessary for several reasons. For example, parameter identifiability in birth-death-sampling models may be a concern for methods simultaneously estimating origination, extinction, and sampling rates, but identifiability can be improved when parameter constraints are available[60]. In addition, we could not plausibly include all known Phanerozoic brittle star taxa in our phylogenetic analysis, but highly incomplete taxon sampling may bias FBD parameter estimation, which in turn may influence macroevolutionary inferences, including ancestor-descendant probabilities. To address these concerns, we conducted several analyses that constrained the fossil sampling rate to reflect values estimated using traditional palaeontological methods (i.e., independent of phylogeny). To obtain an empirically informed prior for the sampling rate, we downloaded all Phanerozoic occurrences of ophiuroid taxa from the Paleobiology Database (paleobiodb.org) and estimated per-interval sampling probabilities using three-timer methods[61,62]. These per-interval sampling probabilities were then converted to rates, and the mean and variance of the Phanerozoic sampling rate distribution were used to specify an empirically informed Beta distribution as a prior on the FBD fossil sampling rate (Supplemental Methods). Furthermore, we also conducted analyses varying clock model priors, such as calibrating the variance parameter of the IGR model and placing empirically informed estimates on the prior specifying the base rate of the clock[63]. All sensitivity analyses recovered identical phylogenetic positions and macroevolutionary inferences for key taxa described here as those recovered by the initial analysis, indicating our results are robust to a wide range of variation in choice of assumptions and prior distributions (Supplementary Methods).

**Plate size analysis**. In order to approximate the body size trends of the Gotland ophiuroids (Fig. 1), we measured the average lateral surface area of the lateral arm plates. We assume that the area of the lateral arm plates is correlated to the size of the arm but not necessarily the diameter of the disc nor the length of the arm. Length and width of the lateral arm plates were measured graphically using the software imageJ. Plate counts, average values and standard deviation for every sampled locality are given in Supplementary Table 1 and Supplementary Data 2. Because Hoburgen yielded only a single lateral arm plate (of very large size compared to the others), we excluded this sample from our analysis.

To assess how robust the results of our time-series analysis are to sampling artifacts and other issues concerning the fidelity of the fossil record, we implemented a significance test using Monte Carlo simulations. We simulated arm plate size data for each sample locality by drawing values from normal distributions having the mean and variance of the empirical data, with the number of fossil occurrences per site equal to the sample size of that site. Note the assumption of normally distributed variation is standard for quantitative characters such as size, which are often influenced by many loci with small effects. However, because the spatial and temporal distribution of fossils can be biased by a number of non-biological factors (e.g., taphonomic processes), we simulated 10,000 hypothetical fossil records and permuted fossil occurrences across times and across localities to investigate a worst-case scenario where patterns of body size change have virtually no meaningful evolutionary signal. For each simulated record, we then examined the patterns of change across the three palaeoenvironmental crisis events (Ireviken, Mulde and Lau Events) to generate null distributions of net change across each event. For each crisis interval $i$, we calculated the change in size ($\Delta_i$) as the difference between mean arm plate area of the oldest and youngest assemblages associated with that event, and compared empirical values of $\Delta_i$ to the null distribution of size changes to calculate p-values associated the observed patterns (Supplementary Fig. 7).

**Statistical methods and reproducibility**. Phylogenetic analyses were conducted in with MrBayes[44]. All Bayesian PP values of 95–99% were interpreted as indicated strong support, ≥90% PP as good support, and ≥50% as positive support. For the Monte Carlo test, p-values <0.05 were considered significant.

## Data availability
All data required to generate the images and interpret the results of this study are included in this published article (and its supplementary information files) and are openly available at https://doi.org/10.5281/zenodo.5619312.

## Code availability
The MrBayes scripts and R code to reproduce the analyses are available at https://doi.org/10.5281/zenodo.5619312.

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

## Acknowledgements

We thank Claude Altmann (Luxembourg) for help with morphometric data collection.

## Author contributions

B.T., M.E.E., L.D.N.-T. and J.L. conceived the study. M.K. collected the samples and picked the specimens. B.T. and L.D.N.-T. scanned and analyzed the specimens and performed the initial Bayesian phylogenetic analysis. D.F.W. performed the quantitative analyses. M.E.E. compiled the geological background. B.T., L.D.N.-T., M.E.E., and J.L. created Figs. 1–3. B.T. and L.D.N.-T. created Supplementary Figs 1–5. D.F.W. created Fig. 3 Supplementary Figs 6, 8. All authors contributed to writing the manuscript.

## Competing interests

The authors declare no competing interests.
