## [Peer Review File · Communications Biology]

Reviewers' comments:

Reviewer #1 (Remarks to the Author):

The nature of the fossil record commonly makes phylogenetic inference speculative, which is why the advent of numerical phylogenetic methods has contributed so much to our understanding of evolutionary history. Yet, results of numerical phylogenetic methods are commonly only inferences that require further testing. In this paper, high-resolution stratigraphic sampling combined with numerical methods demonstrate a robust and highly unexpected conclusion that the bauplan of modern ophiuroids can be traced to a single speciation event. Further, this speciation co-occurs through a Lilliput event during a time of known crisis in the biosphere. I enthusiastically recommend publication of this important manuscript in *Nature Communications Biology*.

Reviewer #2 (Remarks to the Author):

I have reviewed the revision to the Thuy et al manuscript and feel that the revisions have improved the manuscript. It is a very intriguing study that aims to pinpoint the origination of a very important clade of echinoderms to the critical Mulde Event of extinction and recovery. This origination shows a concomitant miniaturization of the taxa suggesting that the origination was through paedomorphosis and a Lilliput Effect. Furthermore, this size reduction resulted in the simplification of the segmented arm plating that characterized the modern brittle star clade. This meshes well with data that is arising from further up section where most of the brittle star taxa are extremely small and include descendants of the species highlighted in this paper. All in all a fine piece of work.

With the improvements in the manuscript I feel that it is ready for publication and I recommend acceptance of the revision for *Nature Communications*.

Specific comments.

Line 19, it would read better if "are considered to" was removed.

Line 55, The sentence here is long and confusing. It might read better as:

Here, we describe two new fossil taxa from Silurian strata on the Island of Gotland, Sweden, placing them in a phylogenetic context. Furthermore, we analyze the stratigraphic succession and associated ophiuroid microfossil assemblages, that collectively consisted of more than 1,300 specimens, (Fig. 1; Supplementary Figs. 1-2), ranging in age from the Telychian (latest Llandovery, about 428 Mya) to Ludfordian (late Ludlow, about 420 Mya)

Line 142, Groove spines

Line 277, would read better as "excellent fossil preservation"

Reviewer #3 (Remarks to the Author):

COMMSBIO-20-1183A is a very interesting paper that presents a plausible case for having recovered a specific example of an ancestor-descendant relationship at the base of a significant biological radiation that corresponds with major environmental perturbations. I am generally sympathetic to the idea that the fossil record includes many direct morphological ancestors, and to the idea that perturbations trigger major radiations (e.g., the radiation of placental mammals in the earliest Palaeocene). However, I have some points to make about the analyses.

First, there are reasonable questions about whether the character matrix (68 characters, less than three per species) is large enough to sustain this degree of detailed interpretation, and specifically about whether there are enough characters to infer that *O. paicei* is the direct ancestor of *M.*

duplantieri, which by convention would mean that it features as few or fewer autapomorphies than one would expect to see due to reversal across the branch leading from ancestor to descendant. If I understand correctly, just two characters differentiate the species (lines 149 and 150), which is suggestive of very little evolution separating them, but does not exclude the possibility that a larger data set would demonstrate autapomorphies. Also, minor differences in character definition or scoring might have led to different conclusions. I'm not suggesting that the authors didn't stare hard enough at the fossils, just that some circumspection is called for.

On lines 371 and 373, something more specific could be said about what kind of assumptions were made. For example, when it comes to rates, what kind of a distribution was assumed, and with what parameters? Readers should not have to go to reference 9 to look this up. This sort of information is given in the following section on the later analysis, but it would help to see details in both sections.

In any case, the results in general mostly hinge on the Bayesian analysis (starting at line 388). I am confident that the authors have done their best to choose reasonable model structures and parameters, but that doesn't overcome the basic problem with this approach: the need to make non-trivial assumptions about shapes of distributions and their parameters. For example, why a log normal distribution of rate variation, and why five bins (lines 400 and 401)? Why a gamma clock model (lines 401 and 402), given that the gamma distribution has two parameters and is therefore poorly constrained (meaning it should present a flat likelihood surface)? Furthermore, the gamma distribution has no biological basis in this context. What do the shape and rate parameters mean in terms of actual evolutionary processes?

I recognise that these queries address a central part of the authors' research programme, and the above is not a full rebuttal. However, I doubt that I am the only researcher who shares these reservations, and specifically who thinks that drawing precise probabilities from model structures that depend on many specific assumptions is not very cautious.

Another issue is that the authors assume speciation, extinction, and sampling probabilities (λ , μ , and ϵ) that vary among time intervals without constraint and are only inferred from the few fossil occurrences that feature in the data set. The remainder of the fossil record in this region is not considered.

Regardless, fossil sampling rates inferred in any way are strongly upwards biased when across-species uniform rates are assumed but species actually vary in their sampling probabilities (which they must do). The reason is that the known species are the most common ones by virtue of rare ones being sifted out. This problem is serious for the current paper because the interpretation hinges on one basic statistical inference: assuming ancestors can be sampled in the first place, and I agree they can be, then they are likely to have actually been sampled because ϵ is high (to simplify the Bayesian logic).

It is also bold to infer sampling rates when virtually all the species are singletons. Almost any ecologist would agree that sampling rates are simply unknown when all species are singletons, because the number of species in the world can't be known. Adding in information from a phylogeny doesn't really salvage the situation because there is no consideration of the size of the species pool or the variation in abundance among species. The larger it is, the higher the likely variance, and so the stronger the sifting effect. Thus, modelling these parameters would seem to be important.

With respect to λ and μ , the authors may not have considered that even when very large data sets that include very many taxa with numerous occurrences are used to infer these rates at the stage or epoch level, sampling variance in rates is quite high, and results differ even between data sets generated using the same data and related methods. See my 2014 paper in Paleobiology for an example. Again, it seems bold to infer turnover rates from a tiny data set of singletons.

On line 418, the authors nominally switch from a pure Bayesian approach depending on phylogenetic data to a discussion of Foote's approach, which originally depended on occurrence data. Statistics are illustrated in Supplementary Figure 6 and suggest poor completeness (peaking

at around 70%), even though the result in this section hinges on the inflated epsilon value discussed above. The authors don't seem to discuss what these low values might mean for the paper's main conclusion. At face value, they indicate that the conclusion is not sufficiently reliable to be pitched strongly.

On line 175, it's not clear why the authors suggest this is a case of budding cladogenesis instead of anagenesis given that the species don't overlap in time. On lines 432 and following, this matter comes up again and the preference for inferring cladogenesis still isn't explained. This may just be an oversight.

The Monte Carlo bootstrapping procedure (lines 194 to 196, Supplementary Figure 8) is explained on lines 451 through 455. A parametric method was used, here assuming normality of the distributions, but it's not clear to me why this was necessary given that a nonparametric bootstrap could have been used instead. The illustrated confidence intervals are extremely small. This might not be a coincidence. Another possibility, not addressed here, is that taphonomic differences among the localities govern the differences among samples.

The distributions shown in Supplementary Figure 9 are also extremely narrow, suggesting the possibility of a technical error or a strong taphonomic overprint. The authors initially state that the fossil occurrence times were "scrambled" as part of the Monte Carlo procedure (line 464), and I would think that this alone would generate broad distributions, not tiny distributions.

I tried to replicate the analysis by eyeballing the values in Supplementary Figure 8, which look like 0.22, 0.29, 0.1, 0.34, 0.11, 0.05, 0.38, 0.26, and 0.12. If you scramble this list and take differences you get a 95% confidence limit of plus or minus 0.29:

```
mm <- c(0.22, 0.29, 0.1, 0.34, 0.11, 0.05, 0.38, 0.26, 0.12)
d <- array()
for (i in 1:10000)
  d <- c(d,diff(sample(mm)))
quantile(d,prob=c(0.025,0.975),na.rm=T)
```

This easily includes all three red lines shown in Supplementary Figure 9.

The difference in results is so large that either the authors or the reviewer must have made a catastrophic mistake.

The authors go on to say (lines 477 through 484) that an entirely different Monte Carlo method was used to construct Supplementary Figure 9, but this passage is unclear to me. Indeed, this page is hard to follow because it suggests that two kinds of tests were performed but only one is tied to a text figure. In any case, I am yet to be convinced that simply scrambling the time series isn't good enough to test the hypotheses.

Another, also quite serious but elementary problem is that in Supplementary Figures 8 and 9 and Supplementary Table 1, units are reported as mm², suggesting there was no log transformation. This raises serious questions about the method used to compute size changes (log ratios should have been used) and to compute standard deviations (these should have been based on logged values, to belabour the point).

We thank the reviewers for their time and constructive comments. We very much appreciate their reviews and we have improved our manuscript accordingly. Below, please find a point-by-point response to the comments raised (*in italics*).

Reviewer #1:

I enthusiastically recommend publication of this important manuscript in Nature Communications Biology.

Reviewer #2:

Specific comments.

Line 19, it would read better if “are considered to” was removed.

Line 55, The sentence here is long and confusing. It might read better as:

Here, we describe two new fossil taxa from Silurian strata on the Island of Gotland, Sweden, placing them in a phylogenetic context. Furthermore, we analyze the stratigraphic succession and associated ophiuroid microfossil assemblages, that collectively consisted of more than 1,300 specimens, (Fig. 1; Supplementary Figs. 1-2), ranging in age from the Telychian (latest Llandovery, about 428 Mya) to Ludfordian (late Ludlow, about 420 Mya)

Line 142, Groove spines

Line 277, would read better as “excellent fossil preservation”

We agree and changed our manuscript accordingly.

Reviewer #3 :

*First, there are reasonable questions about whether the character matrix (68 characters, less than three per species) is large enough to sustain this degree of detailed interpretation, and specifically about whether there are enough characters to infer that *O. paicei* is the direct ancestor of *M. duplantieri*, which by convention would mean that it features as few or fewer autapomorphies than one would expect to see due to reversal across the branch leading from ancestor to descendant. If I understand correctly, just two characters differentiate the species (lines 149 and 150), which is suggestive of very little evolution separating them, but does not exclude the possibility that a larger data set would demonstrate autapomorphies. Also, minor differences in character definition or scoring might have led to different conclusions. I'm not suggesting that the authors didn't stare hard enough at the fossils, just that some circumspection is called for.*

The ratio between characters and taxa in our matrix is typical for morphology-based phylogenies involving fossil invertebrates. There are numerous examples in the literature, including studies published in Nature Communications (e.g., Legg et al. 2013), using character matrices with similar or even lower ratios. Most importantly, however, our character matrix is based on an approach that has been shown to produce rigorous results, which additionally are congruent with molecular data (Thuy

& Stöhr 2016; O'Hara et al. 2014), and highly resilient against missing characters and incompleteness of the fossil record (Thuy & Stöhr 2018). A comparison of our character with others in the literature indicates our matrix is nearly twice the size of matrices commonly used for other clades of marine invertebrates and simulations show the tip-dating phylogenetic methods we use perform well with similarly sized matrices (Barido-Sottani et al., 2020).

Further along these lines, the comment that “*just two characters differentiate the species (lines 149 and 150), which is suggestive of very little evolution separating them*” is refuted by existing literature (e.g., King 2019) and by the nature of our data. In fact, the transition between two successive species is, by definition, a small evolutionary step; otherwise, one could justifiably question the successive nature of the two species involved. The striking conclusion of our study is not the *number* of changes differentiating the two species but their *nature*: the two changes are fundamental to the extent that they diagnose the entire clade including the living ophiuroids. Our results are highly significant because they show the precise moment when major evolutionary change happened at the scale of a speciation event.

On lines 371 and 373, something more specific could be said about what kind of assumptions were made. For example, when it comes to rates, what kind of a distribution was assumed, and with what parameters? Readers should not have to go to reference 9 to look this up. This sort of information is given in the following section on the later analysis, but it would help to see details in both sections.

In any case, the results in general mostly hinge on the Bayesian analysis (starting at line 388). I am confident that the authors have done their best to choose reasonable model structures and parameters, but that doesn't overcome the basic problem with this approach: the need to make non-trivial assumptions about shapes of distributions and their parameters. For example, why a log normal distribution of rate variation, and why five bins (lines 400 and 401)? Why a gamma clock model (lines 401 and 402), given that the gamma distribution has two parameters and is therefore poorly constrained (meaning it should present a flat likelihood surface)? Furthermore, the gamma distribution has no biological basis in this context. What do the shape and rate parameters mean in terms of actual evolutionary processes?

I recognise that these queries address a central part of the authors' research programme, and the above is not a full rebuttal. However, I doubt that I am the only researcher who shares these reservations, and specifically who thinks that drawing precise probabilities from model structures that depend on many specific assumptions is not very cautious.

Another issue is that the authors assume speciation, extinction, and sampling probabilities (λ , μ , and ϵ) that vary among time intervals without constraint and are only inferred from the few fossil occurrences that feature in the data set. The remainder of the fossil record in this region is not considered.

Regardless, fossil sampling rates inferred in any way are strongly upwards biased when across-species uniform rates are assumed but species actually vary in their sampling probabilities (which they must do). The reason is that the known species are the most common ones by virtue of rare ones being sifted out. This problem is serious for the current paper because the interpretation hinges on one basic statistical inference: assuming ancestors can be sampled in the first place, and I agree they can be, then they are likely to have actually been sampled because ϵ is high (to simplify the Bayesian logic).

On line 418, the authors nominally switch from a pure Bayesian approach depending on phylogenetic data to a discussion of Foote's approach, which originally depended on occurrence data. Statistics are illustrated in Supplementary Figure 6 and suggest poor completeness (peaking at around 70%), even though the result in this section hinges on the inflated epsilon value discussed above. The authors don't seem to discuss what these low values might mean for the paper's main conclusion. At face value, they indicate that the conclusion is not sufficiently reliable to be pitched strongly.

The concerns pertaining to the assumptions for the Bayesian analysis and the fossil sampling rates are of a principled nature and have been amply addressed in the literature of the past 5-10 years. While we do acknowledge that there are different opinions as to which approach is the most pertinent, we emphasize the methods we use are entirely conventional in the fields of molecular and paleontological phylogenetics. We feel it is beyond the scope of our contribution to explain the fundamentals of the field.

For example, the use of gamma distributions to model aspects of evolutionary rate heterogeneity goes back to Yang (1994), was discussed in a paleontological context by Wagner and Macrot (2010), and is now commonplace in virtually all phylogenetic analyses and implemented in commonly used software (MrBayes, BEAST2, RevBayes). Notably, our approach to modelling rate heterogeneity among characters and lineages is strongly recommended in the literature (Matze and Wright, 2016; Wright et al., 2016). The claim that there is no biological basis for interpreting hyperparameters is beside the point for fitting estimating evolutionary parameters (see Figures 1-2 below), but also factually incorrect. Gamma distributions arise in probability theory as the sum of exponentially distributed random variables, and therefore naturally arise as a distribution of evolutionary rates when rates are modelled as exponential functions of time. Interpretation of gamma distributions for modelling rate variation among lineages is discussed in the paleontological literature by Wagner and Marcot (2010), Wright (2020) and Wright et al., (2021).

Furthermore, neither lognormal nor gamma distributions for modelling rate variation in phylogenetic studies estimate the shape and rate parameters separately. Similarly, it is incorrect to assert the likelihood surface of the gamma distribution must be flat, and seemingly misunderstands how parameters are estimated using conventional Bayesian approaches. Following the guidelines in Matzke and Wright (2016), we placed a “flat” uniform prior distribution on the hyperparameter of the gamma distribution (Figure 1 below). However, when the prior is multiplied by the likelihood (a function of the data) the posterior probability is demonstrably not flat and the results in a distribution describing how the rates vary among lineages (Figures 1-2 below).

Finally, the doubt expressed « *...that I am the only researcher who shares these reservations...* » is presumptuous given that a major paleontological society sponsored an analytical workshop highlighting these methods during a major academic conference. In fact, the methodological approach we used was the subject of the 2019 analytical methods short course on “Quantitative Methods in Phylogenetic Paleobiology” by the Paleontological Society as part of the Geological Society of America conference, in which one of us (DFW) was an organizer and workshop instructor.

It is also bold to infer sampling rates when virtually all the species are singletons. Almost any ecologist would agree that sampling rates are simply unknown when all species are singletons, because the number of species in the world can't be known. Adding in information from a phylogeny

doesn't really salvage the situation because there is no consideration of the size of the species pool or the variation in abundance among species. The larger it is, the higher the likely variance, and so the stronger the sifting effect. Thus, modelling these parameters would seem to be important.

With respect to lambda and mu, the authors may not have considered that even when very large data sets that include very many taxa with numerous occurrences are used to infer these rates at the stage or epoch level, sampling variance in rates is quite high, and results differ even between data sets generated using the same data and related methods. See my 2014 paper in Paleobiology for an example. Again, it seems bold to infer turnover rates from a tiny data set of singletons.

This is puzzling because it suggests ignorance of common trends in the early evolution of clades. In many major clades (e.g., the earliest tetrapods, birds, echinoids etc.), the stem consists in a nested succession of species, most of which are singletons (e.g., Brusatte et al., 2014). Since our study looks at the origin of a major clade, it is no surprise that we are dealing with a data set of singletons.

On line 175, it's not clear why the authors suggest this is a case of budding cladogenesis instead of anagenesis given that the species don't overlap in time. On lines 432 and following, this matter comes up again and the preference for inferring cladogenesis still isn't explained. This may just be an oversight.

As explained in the text, the species are temporally so close that budding cladogenesis is not unlikely. However, we think that our phrasing in the main text and in the methods section is sufficiently cautious to make clear that this is merely a suggestion. Whether our data represent a case of cladogenesis or anagenesis is not essential for the conclusions of our study.

The Monte Carlo bootstrapping procedure (lines 194 to 196, Supplementary Figure 8) is explained on lines 451 through 455. A parametric method was used, here assuming normality of the distributions, but it's not clear to me why this was necessary given that a nonparametric bootstrap could have been used instead. The illustrated confidence intervals are extremely small. This might not be a coincidence. Another possibility, not addressed here, is that taphonomic differences among the localities govern the differences among samples.

The distributions shown in Supplementary Figure 9 are also extremely narrow, suggesting the possibility of a technical error or a strong taphonomic overprint. The authors initially state that the fossil occurrence times were "scrambled" as part of the Monte Carlo procedure (line 464), and I would think that this alone would generate broad distributions, not tiny distributions.

Apart from the basic methodological issues already addressed further above, the concern raised that there might be strong taphonomic overprint is unwarranted. All the specimens included in our statistics are fully disarticulated and show no sign of sorting, as evidenced by the micropalaeontological composition of the samples. We added an explanation as to why we exclude taphonomic bias.

I tried to replicate the analysis by eyeballing the values in Supplementary Figure 8, which look like 0.22, 0.29, 0.1, 0.34, 0.11, 0.05, 0.38, 0.26, and 0.12. If you scramble this list and take differences you get a 95% confidence limit of plus or minus 0.29:

```
mm <- c(0.22, 0.29, 0.1, 0.34, 0.11, 0.05, 0.38, 0.26, 0.12)
```

```
d <- array()
```

```
for (i in 1:10000)
```

```
  d <- c(d,diff(sample(mm)))
```

```
quantile(d,prob=c(0.025,0.975),na.rm=T)
```

This easily includes all three red lines shown in Supplementary Figure 9.

The difference in results is so large that either the authors or the reviewer must have made a catastrophic mistake.

Since we provided all the data used in our study, there was no need to « eyeball » values, they could have simply been looked up. Also, it is no wonder that Reviewer #3 is unable to reproduce some of the results if they used a different Monte Carlo approach than we did. While it can be argued about which Monte Carlo approach to use best in this case, claiming that our results are irreproducible is blatantly false.

The authors go on to say (lines 477 through 484) that an entirely different Monte Carlo method was used to construct Supplementary Figure 9, but this passage is unclear to me. Indeed, this page is hard to follow because it suggests that two kinds of tests were performed but only one is tied to a text figure. In any case, I am yet to be convinced that simply scrambling the time series isn't good enough to test the hypotheses.

As stated further above, our approaches are neither novel nor unconventional. We provided all relevant references for further reading as well as all data and scripts to replicate the analyses if needed. In contrast, an exhaustive review of the recent literature in order to defend the validity of the approaches we used by far exceeds the scope of our study.

Barido-Sottani, J., van Tiel, N., Hopkins, M.J., Wright, D.F., Stadler, T. and Warnock, R.C.M., 2020. Ignoring fossil age uncertainty leads to inaccurate topology and divergence time estimates using time calibrated tree inference. *Frontiers in Ecology and Evolution*, 8:183 doi: 10.3389/fevo.2020.00183.

Brusatte, S.L., Lloyd, G.T., Wang, S.C. and Norell, M.A., 2014. Gradual assembly of avian body plan culminated in rapid rates of evolution across the dinosaur-bird transition. *Current Biology*, 24(20), 2386-2392.

King, B., 2019. Which morphological characters are influential in a Bayesian phylogenetic analysis? Examples from the earliest osteichthyans. *Biology Letters*, 15(7), <https://doi.org/10.1098/rsbl.2019.0288>.

Legg, D.A., Sutton, M.D. & Edgecombe, G.D., 2013. Arthropod fossil data increase congruence of morphological and molecular phylogenies. *Nature Communications*, 4, 2485.

Matzke, N.J. and Wright, A., 2016. Inferring node dates from tip dates in fossil Canidae: the importance of tree priors. *Biology Letters*, 12(8), p.20160328.

O'Hara, T. D., Hugall, A. F., Thuy, B. & Moussalli, A., 2014. Phylogenomic resolution of the class Ophiuroidea unlocks a global microfossil record. *Current Biology* 24, 1–6.

Thuy, B. & Stöhr, S., 2016. A new morphological phylogeny of the Ophiuroidea (Echinodermata) accords with molecular evidence and renders microfossils accessible for cladistics. *PLoS ONE* 11, e0156140.

Thuy, B. & Stöhr, S., 2018. Unravelling the origin of the basket stars and their allies (Echinodermata, Ophiuroidea, Euryalida). *Scientific Reports* 8, 8493 doi:10.1038/s41598-018-26877-5.

Wagner, P.J. and Marcot, J.D., 2010. Probabilistic phylogenetic inference in the fossil record: Current and future applications. *Paleontological Society Papers*, Volume 16: Quantitative Methods in Paleobiology, 189-211.

Wright, A.M., Lloyd, G.T. and Hillis, D.M., 2016. Modeling character change heterogeneity in phylogenetic analyses of morphology through the use of priors. *Systematic Biology*, 65(4), 602-611.

Wright, D.F., 2019. Clock Models for Character Change, *Paleontological Society Short Course*, "Quantitative Methods in Paleobiology".

Wright, A.M, Wagner, P.J., and Wright, D.F., 2021. Testing character-evolution models in phylogenetic paleobiology: a case study with Cambrian echinoderms. *Elements in Paleontology*, post-review preprint from 2020 available at: [EcoEvoRxiv](https://doi.org/10.32942/osf.io/ykzgz5). doi: 10.32942/osf.io/ykzgz5

Yang, Z., 1994. Maximum likelihood phylogenetic estimation from DNA sequences with variable rates over sites: approximate methods. *Journal of Molecular evolution*, 39(3), 306-314.

Figure 1. (left) The prior probability placed on the hyperprior on the independent gamma rate. (right) The posterior distribution of the gamma hyperprior estimated using Bayesian MCMC. If the likelihood surface were flat, there would be little to no difference between these distributions.

Figure 2. Rates variation among lineages. The histogram shows the distribution of per-branch rate variation corresponding to branch rates in Supplemental Figure 7. The red lines indicate the mean values and 95% credible interval for rates corresponding to the gamma distribution fit to the data. If the likelihood surface for the hyperparameter were flat, then this distribution would appear uniform rather than highly skewed.

Reviewers' comments:

Reviewer #3 (Remarks to the Author):

Thuy et al. have taken a unique approach in their response to my comments. They have overlooked most of them, made diversionary arguments about others, made no substantive changes to their analyses or text, and stated that I am presumptuous and ignorant. The single change made in response to my review was the addition of a passage intended to counter a toss-off comment I made (see below).

The authors have shown that there is nothing I could say that would lead them to revise this manuscript substantively. Thus, I am addressing the following discussion primarily to the editor.

Before beginning, I should note that in my review I chose not to engage with the authors over the general wisdom of their research program. I was only trying to persuade them to clarify their text and improve their analyses. I noted that their work rested on some broad, unrealistic assumptions but that this was not the place for a full rebuttal. Again, I am not interested in a lengthy, detailed debate in press over whether anyone should be carrying out this kind of work. But I am forced here to note exactly why these assumptions render this research program, and this paper, highly questionable. Note that Barido-Sottani et al. (2020) can be consulted to view the state of the art in the body of literature including this manuscript. Some of the key assumptions are:

(1) The preservation rate (epsilon) is constant through time or (in some literature) can be estimated from the phylogeny and temporal distribution of the few fossils in the analysis. The point of the large literature on fossil preservation and diversity estimation (Darwin 1859, chapter 9; Raup 1972, 1976; citations below) is that these rates are definitely highly variable, and that legitimately large data sets are needed to estimate them.

(2) Preservation rates may vary by group, but do not vary spatially, so biogeography is irrelevant and there is no chance that crucial lineages are missing because they evolved outside the study area. But we know that preservation is heavily biased towards North America and Europe (McGowan and Smith 2008; Vilhena and Smith 2013; Close et al. 2016, 2020), and even towards specific regions within those continents (Alroy 1998).

(3) It is easy to measure preservation rates using data sets comprised mostly or entirely of singletons. Inferring sampling probabilities is the bedrock of the community ecology literature on diversity estimation. All ecological estimation methods imply that preservation rates are unmeasurable and approach zero when all species are singletons. Examples include extrapolators such as Chao 1 (Chao 1984) and ACE (Chao and Lee 1992), and curve-fitting methods such as the Michaelis-Menten approach (see Colwell and Coddington 1994 and Keating and Quinn 1998 for reviews: accumulation curves are straight lines when all species are singletons, so richness estimates are non-asymptotic). When all species are singletons, Simpson's D is zero, so its inverse is infinite, implying zero preservation; Fisher's alpha is infinite, implying the same thing; and the log normal can't be fit because only one count class is filled. My point here is that the body of literature including this manuscript attempts to do something ecologists understandably think is impossible.

(4) There is a constant birth-death process, or its variation through time can be inferred from a phylogeny and a few fossils. But we know that there have been numerous adaptive radiations (Simpson 1953) and mass extinctions (Cuvier 1813), and have for years quantified them using legitimately large data sets (e.g., Raup and Sepkoski 1982; Foote 2000, 2005; Alroy 2008; etc.). However, rates are unmeasurable when all species are singletons (see the same literature).

(5) Origination and extinction rates are unequal and unrelated in addition to constant. But we know that these rates are correlated and density-dependent, explaining why diversity curves are logistic (Sepkoski 1978 etc., Alroy 1996, 1998, 2008, 2009; Close et al. 2016; Foote 2000; Foote et al. 2018). Molecular analyses also support this conclusion (Rabosky 2008 and later papers). Thus, rates vary non-randomly through time and are often subequal (e.g., Alroy 2008).

(6) There is no need to directly measure any of these parameters from fossil data because they simply pop out of a character matrix and the temporal distribution of the relevant species, which are usually singletons, as in the current study. Thus, conventional methods of estimating diversity (e.g., half of my papers) and turnover (e.g., Foote 2000; Alroy 2008, 2014) are assumed to be irrelevant. They are not.

In sum, researchers such as Wright are trying to get blood from a stone. There is simply too much non-random, non-uniform, not easily-modelled variation in crucial parameters for them to handle using the minimal data they seek to leverage. Thus, they are likely to overestimate their posterior probabilities, drawing conclusions that are actually unsupported and untestable – as in this paper.

If the authors really want to debate, then, yes, I might seek to publish something like the above. But I have other things to do, so I am happy to let them go their way if they will drop this debate at the current juncture.

I will now explain the situation with respect to the current exchange by listing the substantive points I made and noting the authors' response in each case (or lack of a response).

(1) I began by expressing interest in the paper and sympathy with its goals. This attempt at charity did not elicit a patient and considered reaction.

(2a) The character matrix is insufficient in size for inferring direct ancestry. Here, the authors take a diversionary approach. First, they argue that many other matrices are small. This is not on point, and does not justify the use of a small matrix.

(2b) Second, the authors cite some literature (most of it their own) to suggest that data set size is irrelevant. However, Barido-Sottani et al. (2020) say nothing about inferring ancestry. Thuy and Stöhr (2018) is about basket stars, not methods, and also doesn't discuss this issue. The other references here are also definitely not on the relevant point.

(2c) In any event, the authors here do not address the specific point I made about the fact that they have not shown that a larger data set would not demonstrate non-trivial numbers of autapomorphies. Instead, the authors argue against my toss-off comment that the data are "suggestive of very little evolution separating them" – which I immediately argued against in the same sentence. So, they have quoted me out of context to suggest I advocated a hypothesis we both seem to think is irrelevant.

(3) I noted that "minor differences in character definition or scoring might have led to different conclusions". This point was ignored and no change was made.

(4) I asked for clarification about assumptions and methods at lines 371 to 373. This point was ignored and no change was made.

(5) I asked why a log normal distribution was chosen and why there were five bins. This point was ignored and no change was made.

(6) I also asked for clarification in general about assuming the gamma distribution. I did not say the assumption was flat-out wrong. But where it is not diversionary, much of the rebuttal focuses on this one point. The authors specifically argue that because this assumption is commonly made, it requires no justification in the actual text. Therefore, despite the lengthy discussion in the rebuttal, no changes were made to explain their choice. Again, I raised this issue because I wanted to see clarification in the text, not an argument in a rebuttal. This is another example of the fact that the authors are not interested in making their paper better: they simply want to score points.

(7) I noted specifically that the gamma distribution is poorly constrained and should be flat. The authors divert by saying I said it must be flat, which I did not mean, and by presenting new figures not included in the manuscript. No changes were made to include these figures or the arguments that goes with it. I must emphasise that the disagreement on this one narrow point is not the main problem with the paper: the main problem is that in some cases the analyses are

wrong (see below), and in other cases they rest on debatable assumptions (see above). The authors have done nothing to change them, which is not acceptable.

(7) I suggested caution because not every researcher accepts the authors' personal research program. This was met with an insult (the assertion that I am "presumptuous") and an irrelevant passage about Wright's having organised a short course, which augments the litany of self-citations throughout the rebuttal. By itself, organising a workshop does not validate a research program and raise it to the point where it cannot be questioned. I never took this position about the PS workshop I organised and the methods I presented there: why do the authors? In any case, I did not challenge Wright's qualifications or make ad hominem comments, so this kind of self-justification is not necessary.

(8) As elaborated upon in the first half of this commentary, lambda, mu, and epsilon are estimated here without using any external information, even though such information is abundant. Specifically, background turnover rates and sampling probabilities for marine invertebrates are known from actual data and legitimate statistical methods, and could be used to help shape prior distributions. This point was ignored and no change was made.

(9) When all species are singletons, as here, sampling rates are undefined. Again, see the above. This point was ignored and no change was made. More than anything, it invalidates the paper and the authors' research program.

(10) Much is known about lambda and mu, and what is known specifically is that these rates are hard to fix for exact time intervals (as opposed to the Phanerozoic in general) even with legitimately large data sets and legitimate statistical methods. This point was ignored and no change was made.

(11) The authors have either misunderstood or intentionally altered Foote's approach without explaining themselves clearly. This point was ignored and no change was made.

(12) The modified Foote method suggests that completeness is poor, potentially invalidating the results in general if the statistic is correct. This point was ignored and no change was made.

(13) I asked for one passage to be revised to clarify the anagenesis vs. cladogenesis issue. I charitably suggested that the relevant wording was an oversight. This query was met with an argument instead of a clarification in the text.

(14) A non-parametric bootstrap should have been used. This point was ignored and no change was made.

(15) The illustrated confidence intervals (Supplementary Figure 8) are definitely too small, indicating a basic analytical error. The authors again divert, focusing on the side-issue of whether taphonomy might be important. No substantive change was made, but a defensive passage about taphonomy was added.

(16) The following supplementary figure also shows impossibly small confidence intervals, demonstrating that a major error was committed. The result cannot be replicated by applying an obvious, intuitive, and appropriate bootstrapping method. The authors divert by drawing attention to the availability of their raw data – instead of showing that my estimates were in any way problematic. They divert by asserting that a different method (theirs) would produce a different result. That was actually my point: their method is wrong, and a correct method invalidates their result. They divert again by asserting I have made a "false" point of saying their results are irreproducible. The issue is not reproducibility in a general way: I did reproduce their data. Instead, again, I have redone their analysis and refuted their result.

(17) I requested clarification of the Monte Carlo method. This point was ignored and no change was made.

(18) A crucial analysis is wrong because the data weren't logged. The fundamental need to log

such data is explained to first-year statistics students. This point was ignored and no change was made.

On a new note, the authors condescend to the reviewer by saying "it is beyond the scope of our contribution to explain the fundamentals of the field", as if every reader of a general biology journal is supposed to know all of the fundamentals of this narrow field already. Lack of clarity and detail is a persistent characteristic of this manuscript, and by itself is grounds for rejection. However, I stand by the general point that basic statistical errors were made and that these should have been corrected. Instead, every single substantive specific point I made was ignored or diverted from in the authors' response.

Reviewer #4 (Remarks to the Author):

Review for COMMSBIO-20-1183B-Z

In this manuscript, the authors Thuy et al. study the early origin of brittle stars with the discovery, description and analysis of two new taxa in the Silurian. They have unearthed two new fossil species in the Gotland Island (Sweden) from geological formations of the Silurian. After formal description (by the way, I like the taxon name in honour to Deep Purple), they analyse the external morphology, collect 68 characters, and reconstruct a dated phylogenetic framework using Bayesian inferences and the fossilized-birth-death process. They argue for the discovery of a new pair of temporally consecutive species of brittle stars, which further hides a miniaturization process (likely associated to extinction and environmental changes). They also estimate rate evolution along the phylogeny. Given the early-diverging position of these two taxa before the origin of extant Ophiuroidea, the authors also speculate that a structural simplification of the ophiuroid skeleton through ontogenetic retention of juvenile traits generated the modern brittle star bauplan.

General comments

I am a new reviewer and I have not seen the original submission. However, I have read the reply letter to the comments made during the first round of review. The reviewers were quite positive about the paper, and I found the authors have positively addressed all the comments and suggestions.

Although I am not a specialist of ophiuroids, I really enjoyed reading the manuscript, thus testifying of the importance of this work. Indeed this work addresses a very broad topic in evolution biology: the key origin of an ancient clade in time, space and phenotype. I like the idea and it is well presented in the Introduction. Moreover, the study is well designed and the text is very clear and well written. I found the fossil discoveries quite amazing and I like very much the phylogenetic analyses to place them in a time-calibrated phylogenetic framework using modern approaches like Bayesian tip dating. This has been a pleasure to review this study. Accordingly, I fully support publication of the manuscript in this journal. However, I have some comments I would like the authors address before final acceptance. There are listed as major comments but I don't think they preclude the validity and robustness of the results.

Specific comments

(1) Could you provide a range of ages in each section "Locality and horizon" for each newly described species please? That would be helpful to read this information here.

(2) Perhaps the Discussion will read better if structured with subheadings. For instance, from lines 140 to 152, this is a section focused on the fossil taxa, while the section from lines 153 to 167 is more focused on the phylogenetic position of each taxon. This goes on and on with the speciation mode and evolutionary rates for instance.

(3) Instead of "performed a cladistic analysis" (line 154) I would say "performed a morphological

phylogenetic analysis with 68 characters”.

The term “at the very base” (line 161) is not appropriate. I would recommend saying “early-diverging position” or “sister to”.

The term “comprehensive” (line 162) does not sound the good one here, instead I would say “deep time”. To me comprehensive means that the taxon sampling or the character sampling is very dense, which is not the case here.

The abbreviation “pp” (line 168) should be “PP” as it is all the time.

(4) Lines 172-174: this sentence needs a reference to mentions the study of speciation using the fossil record (e.g. Benton & Pearson 2001 – TREE).

(5) The tip dating MrBayes analyses are a great value of this study. I wonder if the authors can add more results like the divergence times estimation from the tip dating. As it stands, the sentence (lines 175-179) lacks a timeframe in my opinion. Also, I strongly recommend putting Supplementary Figure 7 in the main text. I found it very important to support one of the main conclusions of the study. Just add a proper geological time scale to make it more self-explanatory and appealing.

(6) As a non-expert of the group, I found it difficult to follow why you reach such a conclusion on lines 209 to 211.

(7) I found the concluding paragraph quite bold, especially the first sentence (lines 218-220). Are you for that statement? Can you elaborate more with examples from the literature, especially in the vertebrates?

(8) The reference to MrBayes is outdated. You cite the version 3.1 that does not include the models for molecular clock and fossilized-birth-death. You should cite version 3.2 that is Ronquist et al. (2012 – Syst. Biol.).

(9) I think there is an issue in the reference list for Thuy & Stöhr at line 336. The reference is the number 8 not 9. Check well the reference list.

(10) I am not a fan of looking at other papers to understand what has been done in terms of analyses for the parameters and priors. Could you list the parameters and priors here instead of sending the readers to other studies? We don't know very basic information on the MrBayes analyses like how long the runs were? More importantly, we don't know much about important parameters or priors like the sampling strategy, the molecular clock model... Minor point: the abbreviation ‘mgen’ is useless, remove.

(11) I have downloaded the GitHub repository and checked the files, especially the NEXUS MrBayes file for the tip dating analysis. It is very good to me. I like the comments into brackets to help people who want to reproduce similar analyses on their own group of interest. This is a great initiative. Thanks!

(12) The paragraph (lines 382-404) about the completeness of the fossil record is confusing to me. For instance, I'm not sure to understand what has been used from the MrBayes inferences, and how this has been extracted? Some clarifications are needed.

I am sorry for the picky review, but I do think the study is great and I just want to make the best of your study so that most of the readers fully understand.

Fabien L. Condamine

I would be pleased if the authors choose to contact me for any discussion or information on my review. I was perhaps unclear or wrong on some of the comments.

E-mail: fabien.condamine@gmail.com

Reviewer #5 (Remarks to the Author):

See attached file

Review of "Miniaturization" by Thuy et al

For Communications Biology

I was asked to comment specifically on the methodological aspects of this paper, especially with respect to the concerns raised by reviewer 3.

I will first comment on each of the reviewer's criticisms and then summarize my impression. I give a short phrase in italics and quotes to indicate which section of the reviewer comments I discuss below. The line numbers refer to the revised manuscript.

"... some circumspection is called for"

I agree that the conclusions drawn in the paper are rather bold and that a more cautious interpretation of the results would be preferable. In particular, I think the paper would benefit from a better analysis of the notion of "direct ancestor" and what a statistical analysis of fossil evidence can and cannot tell us about this. I also think the authors need to present an analysis of the sensitivity of their results to alternative priors and other model settings. See my conclusions for concrete advice on both of these points.

The remaining criticisms in this section appear somewhat unfair or, at least, difficult to do anything about. A larger dataset might of course show the presence of autapomorphies, but the analysis must be based on the available data. It is true, as the reviewer states, that minor differences in character definitions or scoring could have led to different conclusions. In pointing out this, however, the reviewer should also have noted that the authors try to address this problem by having two investigators scoring the characters independently. What is lacking in the paper, I think, is a conclusion from this independent scoring exercise. Did it result in perfectly consistent scorings? If not, how many differences were there and how were those differences resolved?

"On lines 371 and 373 ..."

I think the criticism in this section is a mixture of valid points and some rather unspecific complaints about Bayesian statistics, which seem at least partly to be based on misunderstandings. I think the authors are correct in ignoring the latter but I think they missed addressing some of the valid points. In particular, I think the reviewer is right that the authors need to be more specific about many of the details of their analyses. I list the points that appear most important to me below.

It is stated in Material and Methods (line 336) that “We assumed a priori that characters have been subjected to variable rates of evolution, and we accounted for missing data in our matrix.” The authors need to specify what model was used for rate variation across characters, and how they accounted for missing data. There is a de facto standard for treating rate variation across sites/characters, namely the discrete approximation of the gamma distribution with four categories, but it is by no means a given that this is the model that the authors actually used. I am not aware of any standard method of accounting for missing data so the method the authors used for this definitely needs to be explained.

It is claimed on lines 336-337 that “[these assumptions] were found to give the most likely results”. This is not very specific and I think the word “likely” is unfortunate. The likelihood of a result, given a model, is precisely defined, and cannot be used directly to assess whether the assumptions of the model are reasonable. In Bayesian analysis, we typically use Bayes factor tests, which are computed from model likelihoods, to evaluate models. If a rigorous model test is alluded to here, either a Bayes factor test or some form of likelihood test, then please say so. If the sentence refers to some other way of assessing whether the model assumptions are reasonable, then this assessment needs to be described and the word “likely” should preferably not be used.

On line 341, it is stated that “Each of the resulting trees receives a likelihood score and a majority rule consensus is calculated to summarize the variation across the trees”. Each of the two parts of this sentence is true but the word “receives” is confusing and the two parts are not connected in a Bayesian analysis. It would perhaps have been better to say something along the lines of “Trees (and other parameters of the model) are sampled according to their posterior probability (the prior probability times the likelihood score). The tree sample is often summarized by computing a majority rule consensus tree.”

On line 343 it is said that “The likelihood of the tree being true is given as the posterior probability”, which is quite confusing, as the likelihood of a tree is an unscaled value that does not tell you anything about its probability without proper context. I think the authors want to say something like “The posterior probability of a tree is the probability of the tree being correct given the model and the evidence”.

On line 349 the authors talk about “confidence intervals” but in Bayesian statistics we compute “credibility intervals”, which are superficially similar to confidence intervals in some ways but they are based on different concepts and should not be confused with confidence intervals.

Lines 361-362 needs rewording although it is possible to guess what is meant.

On line 363 it is stated that “As described for the undated analysis, we used the Mk model of character evolution.” However, this is not mentioned for the undated analysis as far as I can see.

On lines 373-375, the ESS is given, which is good, but the PSRF (potential scale reduction factor or ‘Gelman and Rubin statistic’) is arguably a better convergence diagnostic and should be given as well.

The reviewer asks why a lognormal distribution was chosen to model rate variation across sites in the dating analysis, and why five discrete categories were chosen to approximate this distribution. The choice of a lognormal distribution is unusual, and could perhaps have been motivated by a comparison with some alternative model, such as the more commonly used gamma model. However, my guess is that the results of the analysis would be fairly similar for both the gamma and the lognormal models.

The number of categories used to approximate the distribution is unlikely to be a concern in either case. Ziheng Yang concluded in the 90’s that four categories are enough to approximate the gamma distribution of rates across sites for most practical purposes. This implies that any number of categories above four would essentially give the same results. This probably also applies to the lognormal distribution, and I see no real need for the authors to justify their choice of five rate categories. Of course, one could check that five and six rate categories give the same results for the lognormal distribution approximation to make sure that this is not a concern.

I think the choice of an independent gamma rates (white noise) model for rate variation across lineages (the relaxed clock) is quite natural, despite the concerns of reviewer 3. There are not that many alternative models of rate variation across lineages that have been explored so far. Most papers use “uncorrelated” models, of which the white noise model is the most elegant from a mathematical standpoint (the others have a spurious dependency on the branching structure of the tree). The uncorrelated models have the big advantage that they are easier to sample from with MCMC than the more biologically plausible autocorrelated models, such as the Thorne-Kishino model. Some biologists argue that the rate of morphological evolution varies more over time than what is accommodated in an autocorrelated model, so this again would suggest that the white noise model is a good choice for this study.

As correctly pointed out by the authors, the distributions used to model rate variation across sites or lineages have just one parameter, since they are scaled to the average rate, not two as claimed by reviewer 3. There is also ample evidence to show that neither the likelihood surface nor the posterior probability distribution for these model components are flat for most datasets, contrary to what is implied by reviewer 3. However, it would have been appropriate to specify the priors and show the posterior distributions for the parameters of the rate variation models. This is not exactly done in Figure 1 and Figure 2 in the rebuttal, which I have a difficult time interpreting. What I would like to see is a figure showing the prior and the posterior for the variance parameter of the white noise model (called ‘igrvarpr’ in MrBayes), and the same for the model of rate variation across sites (‘shapepr’ for the

lognormal or gamma model of rate variation across sites). For the white noise model, I would have liked to see some calibration of the prior to the dataset, for instance using the non-clock – strict-clock branch length correlation technique outlined by Ronquist et al (2012). The prior versus posterior plot would indicate to what extent the posterior may be influenced by the prior.

“It is also bold to infer”

Here, and elsewhere, reviewer 3 discusses the problem with fossil sampling probabilities. I think reviewer 3 has some very good points concerning this. The authors really need to describe the epoch FBD model (fossilized birth death with sampled ancestors and with different parameter values for different strata or epochs) they used and justify the priors they chose for the model parameters. We know that birth-death-sampling models, of which FBD is an example, suffer from identifiability issues, making it impossible to infer birth, death and sampling rates at the same time (Louca & Pennel 2020 and references cited therein). In this study, it appears that birth, death and fossil sampling rates were all inferred from the data, and separately for many different geological strata as well. It would be critical for the authors to show what priors they used, and the sensitivity of their results to alternative specifications of those priors. I would also like to see a discussion of how the fossil sampling priors are related to background knowledge about the fossil sampling probabilities during the studied time interval (this is also pointed out by reviewer 3).

Reviewer 3 points out that variation in the abundance of taxa belonging to different lineages is not accommodated in the model that the authors use, and that this is likely to bias fossil sampling probabilities upwards. It seems quite challenging to address this and I am skeptical that the simplifying assumption of a uniform sampling probability across lineages in the same time interval is a serious problem given the other sources of uncertainty in the analysis.

“On line 175”

Here, reviewer 3 questions the justification for proposing that this is a case of “budding cladogenesis”. I do not think “budding cladogenesis” is a commonly used term; I had not encountered it before reading this paper. I suggest that the term is defined in the paper, or an alternative way of describing the phenomenon is used. However, I would prefer to have this entire argument restructured, as the distinctions between anagenesis, budding cladogenesis (if I guess the meaning of this term correctly) and real cladogenesis is extremely difficult to make and not essential for the conclusions of the paper. See conclusions below.

“The Monte Carlo bootstrapping...”

I agree with reviewer 3 that non-parametric bootstrapping would have been a better choice than parametric bootstrapping. The former technique is less sensitive to model assumptions, such as the values being sampled from normal distributions. The word “scrambled” is also

difficult to understand (line 428). If “scrambled” means that the authors randomly permuted occurrence times to form the reference distribution, then please say so. And also specify if the permutations involved the ages of the sites or of the individual fossils. I would assume the latter given the significance values reported, but it is not obvious and it has important implications. And were occurrence times permuted within zones, or randomly among all sites regardless of zone? In the case of the directional tests, note also that the tests are designed based on observations from data; thus, there is a bias towards significance that may be hard to address.

I also agree with the reviewer that taphonomic bias could be a potential concern. On line 266, the authors say that “the samples show no sign of pre-burial sorting. We therefore exclude any taphonomic bias.” Here I think it would have been relevant to provide some quantitative evidence. How did the authors assess the existence of pre-burial sorting, and how confident are they that there is no such bias? (And by the way, it is “Mulde Tegelbruk” and not “Mulde Tegelbruck” on line 265).

In general, the results of the Monte Carlo tests discussed in this section are difficult to interpret, and it is difficult to match the methods section with the reported results. Where are the results from the directional tests presented? The time axis of Supplementary Figure 8 does not appear to match the time axis for the sites in Supplementary Figure 2. Is there an error in one of them? If fossil specimens are randomly permuted across sites, then could the extreme significance values reported in Supplementary Figure 9 not be caused by local community or habitat effects and not by environmental change? If one were to randomly permute the age of sites instead, would there still be a significant correlation between size decrease and environmental change? If not, then the changes in size over time may be due to random fluctuations or driven by other factors than environmental change.

Conclusions

Overall, I think this is a quite interesting paper. In particular, I am intrigued by the fact that the Bayesian phylogenetic analysis puts so much posterior probability of the two key fossils being sampled ancestors.

For me, there are two main concerns. First, I think the discussion of “direct ancestors” would benefit from more depth. The birth-death-sampling models are quite complicated. An important aspect is scaling. It is like we were looking at the phylogeny with and without a magnifying glass. Without the magnifying glass, it appears that we have few extinct side lineages and thus a low extinction rate. The speciation rate also appears low and the fossil sampling or tip sampling rate appears to be high. However, when we apply the magnifying glass, we see that there are many extinct side lineages that we did not see before. Thus, the speciation and extinction rates now appear to be higher and the sampling rates lower.

What might appear as direct ancestors when we look at the tree without our magnifying glass will almost certainly turn out not to be direct ancestors if we increase the magnification enough. I think this paper presents an analysis of what the tree looks like at low magnification. I am intrigued by the fact that this low-magnification analysis suggests that the two key species are direct ancestors, and I think this will stimulate the interest of many other evolutionary biologists in this system and the apparently sudden evolutionary transition involved. Nevertheless, I would like to see the paper acknowledge this scaling issue and present the results in this context.

In general, it appears to me that the statistical analyses presented in the paper and the assumptions on which they are based are reasonable, and I think some of the criticisms of reviewer 3 are irrelevant or beside the point. Nevertheless, I think that some of the points raised by the reviewer expose real weaknesses in the paper, particularly in the presentation of the models and the analyses. Even more seriously, I think there is a lack of critical assessment of some of the priors and other model assumptions that could potentially have a large influence on the results. Even though this is serious, I do think these issues will be possible to address with some additional work. In summary, the issues I see are:

- The models used in the Bayesian phylogenetic analyses need to be described more precisely to address the questions raised above.
- The presentation of Bayesian phylogenetics needs to be improved, if it is to be included at all (a reference to some other introductory text might be sufficient)
- The Monte Carlo analyses of size trends need to be described better and the results interpreted with more caution and in relation to the randomized reference distributions used.
- The priors and posteriors should be specified for the models of rate variation across sites (in both the non-clock and the dating analyses) and across lineages (the relaxed clock model in the dating analysis). The prior for the relaxed clock model should be calibrated to the dataset. Ideally, one would also examine the effects of varying the priors on the rate variation models and on the age of the root node in the dating analysis.
- The priors used for birth, death and fossil sampling rates for each epoch should be specified and justified with reference to some background information if possible. The sensitivity of the results to variations in these assumptions should be explored and reported in the paper.

References

Louca S, Pennell MW. Extant timetrees are consistent with a myriad of diversification histories. *Nature* 580: 502–5. <https://doi.org/10.1038/s41586-020-2176-1>.

Ronquist F, Klopfstein S, Vilhelmsen S, Schulmeister S, Murray DL, Rasnitsyn AP. 2012. A total-evidence approach to dating with fossils, applied to the early radiation of the Hymenoptera. *Systematic Biology* 61: 973–99. <https://doi.org/10.1093/sysbio/sys058>.

We thank the reviewers for their time and constructive comments. We very much appreciate their suggestions and we have improved our manuscript accordingly. Below, please find a point-by-point response to the comments raised (*in italics*).

Reviewer #4:

(1) Could you provide a range of ages in each section "Locality and horizon" for each newly described species please? That would be helpful to read this information here.

Done.

(2) Perhaps the Discussion will read better if structured with subheadings. For instance, from lines 140 to 152, this is a section focused on the fossil taxa, while the section from lines 153 to 167 is more focused on the phylogenetic position of each taxon. This goes on and on with the speciation mode and evolutionary rates for instance.

We subdivided the Discussion using subheadings.

(3) Instead of "performed a cladistic analysis" (line 154) I would say "performed a morphological phylogenetic analysis with 68 characters".

Changed.

The term "at the very base" (line 161) is not appropriate. I would recommend saying "early-diverging position" or "sister to".

Changed.

The term "comprehensive" (line 162) does not sound the good one here, instead I would say "deep time". To me comprehensive means that the taxon sampling or the character sampling is very dense, which is not the case here.

Changed.

The abbreviation "pp" (line 168) should be "PP" as it is all the time.

Corrected.

(4) Lines 172-174: this sentence needs a reference to mentions the study of speciation using the fossil record (e.g. Benton & Pearson 2001 – TREE).

Reference added.

(5) The tip dating MrBayes analyses are a great value of this study. I wonder if the authors can add more results like the divergence times estimation from the tip dating. As it stands, the sentence (lines 175-179) lacks a timeframe in my opinion.

Thank you for this comment. We have added components describing the divergence-time estimation and a figure (Fig. 4) showing its results directly into the main text.

Also, I strongly recommend putting Supplementary Figure 7 in the main text. I found it very important to support one of the main conclusions of the study. Just add a proper geological time scale to make it more self-explanatory and appealing.

We agree and transformed Supplementary Figure 7 according to your suggestions, and moved it to the main text.

(6) As a non-expert of the group, I found it difficult to follow why you reach such a conclusion on lines 209 to 211.

We modified this section to better explain our conclusion.

(7) I found the concluding paragraph quite bold, especially the first sentence (lines 218-220). Are you for that statement? Can you elaborate more with examples from the literature, especially in the vertebrates?

We tuned down this statement to make it less bold. For the sake of conciseness, we refrain from referring to other examples from the literature and instead state that we present « ... one of the exceptionally rare cases where the origin of a major animal clade can be pinpointed precisely in terms of timing, location and driving paleoecological mechanisms ... ».

(8) The reference to MrBayes is outdated. You cite the version 3.1 that does not include the models for molecular clock and fossilized-birth-death. You should cite version 3.2 that is Ronquist et al. (2012 – Syst. Biol.).

Corrected.

(9) I think there is an issue in the reference list for Thuy & Stöhr at line 336. The reference is the number 8 not 9. Check well the reference list.

You are right. We corrected the reference number.

(10) I am not a fan of looking at other papers to understand what has been done in terms of analyses for the parameters and priors. Could you list the parameters and priors here instead of sending the

readers to other studies? We don't know very basic information on the MrBayes analyses like how long the runs were? More importantly, we don't know much about important parameters or priors like the sampling strategy, the molecular clock model... Minor point: the abbreviation 'mgen' is useless, remove.

We added the relevant details and removed ,mgen'.

We agree this is very useful to add. Descriptions of the phylogenetic methods, including what the reviewer has asked for here and much more, are now included in the methods section of the main text. In addition, a full description of the precise priors used and their rationale has been included in the Supplementary Methods.

(11) I have downloaded the GitHub repository and checked the files, especially the NEXUS MrBayes file for the tip dating analysis. It is very good to me. I like the comments into brackets to help people who want to reproduce similar analyses on their own group of interest. This is a great initiative. Thanks!

Thank you for this constructive comment.

(12) The paragraph (lines 382-404) about the completeness of the fossil record is confusing to me. For instance, I'm not sure to understand what has been used from the MrBayes inferences, and how this has been extracted? Some clarifications are needed.

Thank you for pointing out this was not clear. The text has been modified to make it clearer that the parameter values used are from the output of the tip-dated analysis.

Reviewer #5:

The remaining criticisms in this section appear somewhat unfair or, at least, difficult to do anything about. A larger dataset might of course show the presence of autapomorphies, but the analysis must be based on the available data. It is true, as the reviewer states, that minor differences in character definitions or scoring could have led to different conclusions. In pointing out this, however, the reviewer should also have noted that the authors try to address this problem by having two investigators scoring the characters independently. What is lacking in the paper, I think, is a conclusion from this independent scoring exercise. Did it result in perfectly consistent scorings? If not, how many differences were there and how were those differences resolved?

We provided details on how the single case of scoring inconsistency was addressed.

It is stated in Material and Methods (line 336) that "We assumed a priori that characters have been subjected to variable rates of evolution, and we accounted for missing data in our matrix." The authors need to specify what model was used for rate variation across characters, and how they accounted for missing data. There is a de facto standard for treating rate variation across sites/characters, namely the discrete approximation of the gamma distribution with four categories, but it is by no means a given that this is the model that the authors actually used. I am not aware of

any standard method of accounting for missing data so the method the authors used for this definitely needs to be explained.

We added the relevant details to the text and is more fully explained in the Supplementary Methods.

It is claimed on lines 336-337 that “[these assumptions] were found to give the most likely results”. This is not very specific and I think the word “likely” is unfortunate. The likelihood of a result, given a model, is precisely defined, and cannot be used directly to assess whether the assumptions of the model are reasonable. In Bayesian analysis, we typically use Bayes factor tests, which are computed from model likelihoods, to evaluate models. If a rigorous model test is alluded to here, either a Bayes factor test or some form of likelihood test, then please say so. If the sentence refers to some other way of assessing whether the model assumptions are reasonable, then this assessment needs to be described and the word “likely” should preferably not be used.

Thank you for pointing out our unintentional use of confusing text. The lines in question have been replaced by a more detailed account of the assumptions used.

On line 341, it is stated that “Each of the resulting trees receives a likelihood score and a majority rule consensus is calculated to summarize the variation across the trees”. Each of the two parts of this sentence is true but the word “receives” is confusing and the two parts are not connected in a Bayesian analysis. It would perhaps have been better to say something along the lines of “Trees (and other parameters of the model) are sampled according to their posterior probability (the prior probability times the likelihood score). The tree sample is often summarized by computing a majority rule consensus tree.”

Corrected.

On line 343 it is said that “The likelihood of the tree being true is given as the posterior probability”, which is quite confusing, as the likelihood of a tree is an unscaled value that does not tell you anything about its probability without proper context. I think the authors want to say something like “The posterior probability of a tree is the probability of the tree being correct given the model and the evidence”.

Corrected.

On line 349 the authors talk about “confidence intervals” but in Bayesian statistics we compute “credibility intervals”, which are superficially similar to confidence intervals in some ways but they are based on different concepts and should not be confused with confidence intervals.

Corrected.

Lines 361-362 needs rewording although it is possible to guess what is meant.

This sentence has been removed and is replaced by a discussion of sensitivity analyses in the main text and more fully described in the Supplementary Methods and Supplementary Table 1.

On line 363 it is stated that "As described for the undated analysis, we used the Mk model of character evolution." However, this is not mentioned for the undated analysis as far as I can see. On lines 373-375, the ESS is given, which is good, but the PSRF (potential scale reduction factor or 'Gelman and Rubin statistic') is arguably a better convergence diagnostic and should be given as well.

The details of the undated analysis have been added. Also, we have included PSRF value (~1.00 for all parameters).

The reviewer asks why a lognormal distribution was chosen to model rate variation across sites in the dating analysis, and why five discrete categories were chosen to approximate this distribution. The choice of a lognormal distribution is unusual, and could perhaps have been motivated by a comparison with some alternative model, such as the more commonly used gamma model. However, my guess is that the results of the analysis would be fairly similar for both the gamma and the lognormal models.

The number of categories used to approximate the distribution is unlikely to be a concern in either case. Ziheng Yang concluded in the 90's that four categories are enough to approximate the gamma distribution of rates across sites for most practical purposes. This implies that any number of categories above four would essentially give the same results. This probably also applies to the lognormal distribution, and I see no real need for the authors to justify their choice of five rate categories. Of course, one could check that five and six rate categories give the same results for the lognormal distribution approximation to make sure that this is not a concern.

We agree that lognormal distributions are less commonly used. We have added a discussion in the Supplementary Methods section but removed its discussion in the main text. The rationale was based on a study by Peter Wagner (2012) that found rate variation in morphologic characters have a slightly better fit to lognormal distributions over gamma. The reviewer is correct that the fit is only marginal and likely has little influence on downstream results. To our knowledge, the number of rate categories necessary for gamma vs. lognormal distributions has only really been discussed by Harrison and Larsson (2015), which suggest lognormal distributions marginally benefit from additional rate categories.

References:

Harrison, L.B. and Larsson, H.C., 2015. Among-character rate variation distributions in phylogenetic analysis of discrete morphological characters. *Systematic Biology*, 64(2), pp.307-324.

Wagner, P.J., 2012. Modelling rate distributions using character compatibility: implications for morphological evolution among fossil invertebrates. *Biology Letters*, 8(1), pp.143-146.

I think the choice of an independent gamma rates (white noise) model for rate variation across lineages (the relaxed clock) is quite natural, despite the concerns of reviewer 3. There are not that many alternative models of rate variation across lineages that have been explored so far. Most

papers use “uncorrelated” models, of which the white noise model is the most elegant from a mathematical standpoint (the others have a spurious dependency on the branching structure of the tree). The uncorrelated models have the big advantage that they are easier to sample from with MCMC than the more biologically plausible autocorrelated models, such as the Thorne-Kishino model. Some biologists argue that the rate of morphological evolution varies more over time than what is accommodated in an autocorrelated model, so this again would suggest that the white noise model is a good choice for this study.

Thank you for this comment. We agree the IGR clock model is appropriate for these data and the question at hand for a variety of methodological and biological reasons.

As correctly pointed out by the authors, the distributions used to model rate variation across sites or lineages have just one parameter, since they are scaled to the average rate, not two as claimed by reviewer 3. There is also ample evidence to show that neither the likelihood surface nor the posterior probability distribution for these model components are flat for most datasets, contrary to what is implied by reviewer 3. However, it would have been appropriate to specify the priors and show the posterior distributions for the parameters of the rate variation models. This is not exactly done in Figure 1 and Figure 2 in the rebuttal, which I have a difficult time interpreting. What I would like to see is a figure showing the prior and the posterior for the variance parameter of the white noise model (called ‘igrvarpr’ in MrBayes), and the same for the model of rate variation across sites (‘shapepr’ for the lognormal or gamma model of rate variation across sites). For the white noise model, I would have liked to see some calibration of the prior to the dataset, for instance using the non-clock – strict-clock branch length correlation technique outlined by Ronquist et al (2012). The prior versus posterior plot would indicate to what extent the posterior may be influenced by the prior.

Thank you for confirming only a single parameter is used to model rate variation.

The prior for the IGR variance parameter used in the initial analysis was:

IGRclockvarpr = Uniform(0.001, 200);

Rev. Figure 1. (left) Posterior distribution of the IGR variance parameter; (right) posterior distribution of the sigma parameter for the lognormal distribution modeling among site variation. Note this distribution is the variance parameter, not the shape of the among site rate distribution itself.

A discussion of why this prior was selected is now included in the Supplementary Methods. In addition, we also discuss a series of sensitivity analyses that assess variation in this assumption, including an analysis that uses the method of Ronquist et al. (2012) to calibrate prior on the IGR variance parameter and the base rate of the clock.

Here, and elsewhere, reviewer 3 discusses the problem with fossil sampling probabilities. I think reviewer 3 has some very good points concerning this. The authors really need to describe the epoch FBD model (fossilized birth death with sampled ancestors and with different parameter values for different strata or epochs) they used and justify the priors they chose for the model parameters. We know that birth-death-sampling models, of which FBD is an example, suffer from identifiability issues, making it impossible to infer birth, death and sampling rates at the same time (Louca & Pennell 2020 and references cited therein). In this study, it appears that birth, death and fossil sampling rates were all inferred from the data, and separately for many different geological strata as well. It would be critical for the authors to show what priors they used, and the sensitivity of their results to alternative specifications of those priors. I would also like to see a discussion of how the fossil sampling priors are related to background knowledge about the fossil sampling probabilities during the studied time interval (this is also pointed out by reviewer 3).

Thank you for this comment. We agree this is a good point. We have more fully described the FBD priors we initially used in the Supplementary Methods. In addition, we conducted several additional analyses using empirical data from the Paleobiology Database to empirically constrain and/or inform fossil sampling rates. We have added an additional section in the text describing this method, and a full description is provided in the Supplementary Methods.

Reviewer 3 points out that variation in the abundance of taxa belonging to different lineages is not accommodated in the model that the authors use, and that this is likely to bias fossil sampling probabilities upwards. It seems quite challenging to address this and I am skeptical that the simplifying assumption of a uniform sampling probability across lineages in the same time interval is a serious problem given the other sources of uncertainty in the analysis.

We agree this would be challenging to address. We are not aware of any phylogenetic model that accounts for differences in the relative abundance of fossil taxa, but agree it is a topic that warrants future investigation. One major difficulty that comes to mind involves how “abundances” in the fossil record reflect both fossil sampling and differences in species ecology, which would be very difficult to tease apart with even really good fossil records. It is an interesting point, but we agree with the reviewer that our assumption of uniform probabilities within intervals, but not between intervals, is unlikely to be a serious issue for our study.

Here, reviewer 3 questions the justification for proposing that this is a case of “budding cladogenesis”. I do not think “budding cladogenesis” is a commonly used term; I had not encountered it before reading this paper. I suggest that the term is defined in the paper, or an alternative way of describing the phenomenon is used. However, I would prefer to have this entire argument

restructured, as the distinctions between anagenesis, budding cladogenesis (if I guess the meaning of this term correctly) and real cladogenesis is extremely difficult to make and not essential for the conclusions of the paper. See conclusions below.

We have added a citation and attempted to clarify our meaning. It is a common term in the paleontological literature, and important for discussions that relate to ancestor-descendant relationships and patterns of speciation in the fossil record. We agree the specific details of whether the pattern we document represents anagenesis vs. either “budding” or “bifurcating” cladogenesis is not terribly important with respect to our conclusions.

I agree with reviewer 3 that non-parametric bootstrapping would have been a better choice than parametric bootstrapping. The former technique is less sensitive to model assumptions, such as the values being sampled from normal distributions. The word “scrambled” is also difficult to understand (line 428). If “scrambled” means that the authors randomly permuted occurrence times to form the reference distribution, then please say so. And also specify if the permutations involved the ages of the sites or of the individual fossils. I would assume the latter given the significance values reported, but it is not obvious and it has important implications. And were occurrence times permuted within zones, or randomly among all sites regardless of zone? In the case of the directional tests, note also that the tests are designed based on observations from data; thus, there is a bias towards significance that may be hard to address.

We have removed discussions about non-parametric bootstrapping and replaced it with a more accurate description of the methods used.

I also agree with the reviewer that taphonomic bias could be a potential concern. On line 266, the authors say that “the samples show no sign of pre-burial sorting. We therefore exclude any taphonomic bias.” Here I think it would have been relevant to provide some quantitative evidence. How did the authors assess the existence of pre-burial sorting, and how confident are they that there is no such bias?

We semi-quantitatively assessed relative plate type abundances of various echinoderm groups in the samples compared to their expected anatomical frequencies in the respective skeletons to see if particular types of plates are under- or overrepresented. All plate types, irrespective of size and shape, were as abundant as one would expect when echinoderm skeletons fall apart without hydrodynamic disturbance, thus precluding substantial pre-burial sorting. We expanded the methods section to explain this approach.

(And by the way, it is “Mulde Tegelbruk” and not “Mulde Tegelbruck” on line 265).

Corrected.

In general, the results of the Monte Carlo tests discussed in this section are difficult to interpret, and it is difficult to match the methods section with the reported results. Where are the results from the directional tests presented? The time axis of Supplementary Figure 8 does not appear to match the time axis for the sites in Supplementary Figure 2. Is there an error in one of them? If fossil specimens are randomly permuted across sites, then could the extreme significance values reported in

Supplementary Figure 9 not be caused by local community or habitat effects and not by environmental change? If one were to randomly permute the age of sites instead, would there still be a significant correlation between size decrease and environmental change? If not, then the changes in size over time may be due to random fluctuations or driven by other factors than environmental change.

We have eliminated some of the more confusing Monte Carlo analyses from the manuscript entirely, as our results do not hinge on their results (or our ability to describe them!). Instead, we have tried to more adequately explain how the remaining Monte Carlo test was conducted in the main text.

Conclusions

Overall, I think this is a quite interesting paper. In particular, I am intrigued by the fact that the Bayesian phylogenetic analysis puts so much posterior probability of the two key fossils being sampled ancestors.

Thank you. We also think this is interesting!

For me, there are two main concerns. First, I think the discussion of “direct ancestors” would benefit from more depth. The birth-death-sampling models are quite complicated. An important aspect is scaling. It is like we were looking at the phylogeny with and without a magnifying glass. Without the magnifying glass, it appears that we have few extinct side lineages and thus a low extinction rate. The speciation rate also appears low and the fossil sampling or tip sampling rate appears to be high. However, when we apply the magnifying glass, we see that there are many extinct side lineages that we did not see before. Thus, the speciation and extinction rates now appear to be higher and the sampling rates lower. What might appear as direct ancestors when we look at the tree without our magnifying glass will almost certainly turn out not to be direct ancestors if we increase the magnification enough. I think this paper presents an analysis of what the tree looks like at low magnification. I am intrigued by the fact that this low-magnification analysis suggests that the two key species are direct ancestors, and I think this will stimulate the interest of many other evolutionary biologists in this system and the apparently sudden evolutionary transition involved. Nevertheless, I would like to see the paper acknowledge this scaling issue and present the results in this context.

We agree with the reviewer and expanded the ancestor-descendent discussion to take into account the scaling effect.

In general, it appears to me that the statistical analyses presented in the paper and the assumptions on which they are based are reasonable, and I think some of the criticisms of reviewer 3 are irrelevant or beside the point. Nevertheless, I think that some of the points raised by the reviewer expose real weaknesses in the paper, particularly in the presentation of the models and the analyses. Even more seriously, I think there is a lack of critical assessment of some of the priors and other model assumptions that could potentially have a large influence on the results. Even though this is serious, I do think these issues will be possible to address with some additional work. In summary, the issues I see are:

- The models used in the Bayesian phylogenetic analyses need to be described more precisely to address the questions raised above.

Done.

- The presentation of Bayesian phylogenetics needs to be improved, if it is to be included at all (a reference to some other introductory text might be sufficient)

We included a more detailed description of the specific Bayesian methods and prior assumptions in the main text and Supplementary Methods.

- The Monte Carlo analyses of size trends need to be described better and the results interpreted with more caution and in relation to the randomized reference distributions used.

We have re-written this section to help improve its quality and clarity.

- The priors and posteriors should be specified for the models of rate variation across sites (in both the non-clock and the dating analyses) and across lineages (the relaxed clock model in the dating analysis). The prior for the relaxed clock model should be calibrated to the dataset. Ideally, one would also examine the effects of varying the priors on the rate variation models and on the age of the root node in the dating analysis.

We have added these details to the main text and Supplementary Methods. In addition, we have followed the reviewer's suggestion to examine the effects and many alternative priors by conducting six sensitivity analyses described in the Supplementary Methods.

- The priors used for birth, death and fossil sampling rates for each epoch should be specified and justified with reference to some background information if possible. The sensitivity of the results to variations in these assumptions should be explored and reported in the paper.

We have followed the reviewers suggest to more fully justify our choice of priors and their sensitivity. Please see the sections on sensitivity analyses in the main text, as well as the Supplementary Methods section on how we obtained empirical estimates of fossil sampling to specify priors for FBD analyses.

REVIEWERS' COMMENTS:

Reviewer #5 (Remarks to the Author):

I am happy with the revised version of this manuscript; the authors have done a good job. The revised version addresses most of my concerns; a few minor exceptions are detailed below. I am sure they can be handled without any further assistance from me.

Remaining comments:

L489ff: "Trees (and other parameters of the model) are sampled according to their posterior probability (the log-prior probability plus the log-likelihood score)." The original text was not correct and the revised text is not correct either. The log-prior probability plus the log-likelihood score gives the log of the posterior probability, and not the posterior probability. And you have to normalize the values to get a probability distribution. And these are marginal probabilities for each parameter, so deriving the posterior probability typically involves summation and integration. This is standard Bayesian inference, though, so I do not think it is necessary to go into this amount of detail. I suggest shortening this sentence to "Trees (and other parameters of the model) are sampled from the corresponding marginal posterior probability distributions."

L189ff: I used a magnifying glass as an analogy to explain the scaling issues, but if that analogy is not used, then "resolution" is probably a better word than "magnification".

L209: Note that the conclusion here runs opposite to the preceding text, which states (correctly I think) that direct ancestors are more likely to appear in low-resolution reconstructions than in high-resolution ones. Thus, the current study is an exception only to the extent that it is a comparably high-resolution study of past evolutionary lineages. This suggests that "in spite of the low magnification" should be replaced with "in spite of the (comparatively) high resolution".

L211 and L646: "budding cladogenesis" still used without explanation. Just adding a few lines to explain this term would be appreciated by many readers, I am sure.

L726: "then examined in patterns" -> "then examined the patterns"

We thank the reviewer for the time and constructive comments. We very much appreciate their suggestions and we have improved our manuscript accordingly. Below, please find a point-by-point response to the comments raised (*in italics*).

Reviewer #5:

Remaining comments:

L489ff: "Trees (and other parameters of the model) are sampled according to their posterior probability (the log-prior probability plus the log-likelihood score)." The original text was not correct and the revised text is not correct either. The log-prior probability plus the log-likelihood score gives the log of the posterior probability, and not the posterior probability. And you have to normalize the values to get a probability distribution. And these are marginal probabilities for each parameter, so deriving the posterior probability typically involves summation and integration. This is standard Bayesian inference, though, so I do not think it is necessary to go into this amount of detail. I suggest shortening this sentence to "Trees (and other parameters of the model) are sampled from the corresponding marginal posterior probability distributions."

Done.

L189ff: I used a magnifying glass as an analogy to explain the scaling issues, but if that analogy is not used, then "resolution" is probably a better word than "magnification".

Corrected.

L209: Note that the conclusion here runs opposite to the preceding text, which states (correctly I think) that direct ancestors are more likely to appear in low-resolution reconstructions than in high-resolution ones. Thus, the current study is an exception only to the extent that it is a comparably high-resolution study of past evolutionary lineages. This suggests that "in spite of the low magnification" should be replaced with "in spite of the (comparatively) high resolution".

Corrected.

L211 and L646: "budding cladogenesis" still used without explanation. Just adding a few lines to explain this term would be appreciated by many readers, I am sure.

We added a few lines to explain "budding cladogenesis".

L726: "then examined in patterns" -> "then examined the patterns"

Corrected.